# Nanoarchitecture factors of solid electrolyte interphase formation via 3D nano-rheology microscopy and surface force-distance spectroscopy

Yue Chen ®[1,2,3] ✉, Wenkai Wu ®[4], Sergio Gonzalez-Munoz ®[1], Leonardo Forcieri[1], Charlie Wells ®[1], Samuel P. Jarvis ®[1], Fangling Wu ®[1], Robert Young ®[1], Avishek Dey[3,5], Mark Isaacs[6], Mangayarkarasi Nagarathinam[7], Robert G. Palgrave ®[5], Nuria Tapia-Ruiz ®[3,8] & Oleg V. Kolosov ®[1] ✉

The solid electrolyte interphase in rechargeable Li-ion batteries, its dynamics and, significantly, its nanoscale structure and composition, hold clues to high-performing and safe energy storage. Unfortunately, knowledge of solid electrolyte interphase formation is limited due to the lack of in situ nano-characterization tools for probing solid-liquid interfaces. Here, we link electrochemical atomic force microscopy, three-dimensional nano-rheology microscopy and surface force-distance spectroscopy, to study, in situ and operando, the dynamic formation of the solid electrolyte interphase starting from a few 0.1 nm thick electrical double layer to the full three-dimensional nanostructured solid electrolyte interphase on the typical graphite basal and edge planes in a Li-ion battery negative electrode. By probing the arrangement of solvent molecules and ions within the electric double layer and quantifying the three-dimensional mechanical property distribution of organic and inorganic components in the as-formed solid electrolyte interphase layer, we reveal the nanoarchitecture factors and atomistic picture of initial solid electrolyte interphase formation on graphite-based negative electrodes in strongly and weakly solvating electrolytes.

The nanoarchitecture of solid-liquid interfaces, in which charge and/or mass transfer and energy conversion predominantly occurs, determines properties of interfacial reactions that, in turn, define energy conversion efficiency, dynamics of species transfer, and even information transfer in many important physical, chemical and biological processes, such as heterogeneous reactions in electrocatalysts, surface passivation in batteries, antigen-antibody interactions and bioelectrical information transmission cross cells. Among them, the solid electrolyte interphase (SEI)[1–5], a passivation layer formed on the battery electrode-electrolyte interface, affects a host of chemical,

[1]Department of Physics, Lancaster University, Lancaster LA1 4YB, UK. [2]Fujian Normal University, College of Physics and Energy, 350117 Fuzhou, China. [3]The Faraday Institution, Quad One, Harwell Science and Innovation Campus, OX11 0RA Didcot, UK. [4]College of Engineering, Swansea University, Bay Campus, Fabian Way, Swansea SA18EN, UK. [5]EPSRC National Facility for XPS (HarwellXPS), Research Complex at Harwell (RCaH), Harwell, Didcot, Oxfordshire OX11 OFA, UK. [6]Department of Chemistry, Lancaster University, Lancaster LA1 4YB, UK. [7]Department of Chemistry, University College London, 20 Gordon Street, London WC1H 0AJ, UK. [8]Department of Chemistry, Molecular Sciences Research Hub, White City Campus, Imperial College London, London W12 0BZ, UK. ✉e-mail: yuechen@fjnu.edu.cn; o.kolosov@lancaster.ac.uk

morphological, electrical, and mechanical properties that pre-determine key battery properties, including initial coulombic efficiency, reversible capacity, cycling and rate stability, and, very importantly, safety. Therefore, multiple research groups worldwide focused on obtaining in-depth SEI knowledge, such as formation mechanisms[2], composition[3], micro/nano-structural[4], and chemical evolution[5]. Among these research topics, understanding the SEI formation mechanism is still most challenging and least explored area.

The challenge of understanding SEI formations is exacerbated by the diversity of interfacial nanoarchitectures of the anode active material[6] and electrical double layer (EDL)[7] on which numerous surface reactions taking place in interlinked sequences. For example, the most common intercalation-based graphite anode has highly anisotropic surface nanoarchitectures with the graphite basal planes mainly consisting of $sp^2$ hybridized carbon atom planes, characterized by atomic flatness, low defect density[8]; while the graphite edge planes are lithium-ion intercalation-active and contain $sp^3$ sites and dangling bonds, owing to the abrupt lattice termination. Therefore, the SEI formation mechanism should be analyzed separately between the graphite basal planes and edge planes[9,10]. More importantly, on the electrolyte side, electrolyte bulk and interfacial solvation structures (EDL nanoarchitectures) that predetermine the electrolyte supramolecular interactions, including metal coordination, van der Waals forces, and electrostatic effects, play significant roles in charge/ion transport and the initial SEI formation and have attracted enormous attention in research and industrial community. It was reported that modulating the electrolyte salt concentration, anion species, non-solvent additives, and co-solvent, various new electrolytes, including solvent-in-salt electrolyte (SIS)[11], water-in-salt electrolyte (WIS)[12] and hydrate-melt electrolyte[13], low-coordination-number solvents electrolyte (LCNSs)[14], localized high concentration electrolyte (LHCE)[15,16] and ultralow-concentration electrolyte (ULCE)[17] etc., can generate a mechanically and electrochemically robust electrolyte-electrode interface significantly improving lithium/sodium ion battery performance[18,19]. The highly concentrated electrolyte system, such as SIS, WIS, and Hydrate-melt electrolytes all promote interface compatibility by facilitating the anion preferential decomposition and reducing the solvent reductive decomposition inside the inner Helmholtz layer. Similarly, many LCNSs and LHCE electrolytes also achieve electrochemically stable interfaces and excellent cycle stability by introducing low-polarity[20] and non-solvation co-solvent to modulate electrolyte solvation structure without increasing the overall electrolyte concentration. More recently studies indicated that the solvents and anions in the primary cation solvation sheaths dominate the electrolyte reduction reaction within the EDL[15,21], and therefore the SEI structure is affected (and, hence, can be beneficially tuned) by the arrangement of solvents/anions inside the EDL via adjusting salt concentration, co-solvent component, and polarization electrical field[7,22]. However, although spectroscopic techniques, such as Raman Spectroscopy, have been used to study the EDL molecular structure near gold electrode surface[22,23], the real space molecular arrangements of electrolyte solvents/anions inside the EDL of carbon-based anode, and especially the evolution of these anion/solvent arrangement toward the structured SEI formation mainly remain elusive[24–26].

Another significant obstacle in studying SEI is due to the fact that its formation involves dynamic processes on a solid-liquid interface, including, but not only, Helmholtz layer formation[27], cation desolvation[28] and accumulation of decomposition products on the electrode surface, which cannot be effectively detected during postmortem studies. Characterizing the dynamic nanostructures of the EDL and SEI under operando conditions in a commercial organic electrolyte, currently remains a major scientific challenge[29–31]. State-of-the-art operando scanning probe microscopy (SPM) techniques have provided significant inroads into the characterization of dynamic SEI formation, through the morphological observations of electrolyte decomposition on anode-electrolyte interface and revealed in-depth insights into its structure[2,32–34]. Unfortunately, traditional SPM observations are limited to the SEI surface properties, with details of 3D internal structure[35] of SEI and nanoarchitecture of EDL generally inaccessible. The 3D subsurface distribution of organic and inorganic phases inside the SEI layer, whether it is a multilayer model[4] or mosaic-type micro-structure model[36], is still under debate[37]. Extending the traditional approach-retract SPM force spectroscopy[35,37–39] to dynamic modulation force spectroscopy[40] that map frequency-dependent nanomechanical properties, the key parameter enabling spatially resolved differentiation of the organic and inorganic SEI components, can further uncover the fine structure of SEI layer and reveal the complex puzzle of SEI structure model. It is worth noting that the soft "gel" part of SEI that was recently reported by surface force apparatus[41] measurements and Cryo-Transmission electronic microscopy (TEM)[37] visualization. This quasi-solid state SEI layer structure enriches the understanding of the complicated SEI structures observed so far, beyond the classic mosaic and multilayer SEI structures[42,43]. Although the gel/quasi-solid structure in the electrochemical solid-liquid interface has raised enormous attention and been reported in situ liquid-phase TEM[44], a native characterization environment not requiring high vacuum can be further achieved by EC-AFM.

Apart from traditional EC-AFM[2,10,45–47], various liquid environment in situ/operando AFM techniques, including scanning electrochemical cell microscopy (SECCM)[48,49], scanning electrochemical microscopy (SECM)[50], tip-enhanced Raman spectroscopy (TERS) AFM[51], mechanical cyclic voltammetry (mCV) AFM[52] and EC-3D-AFM[53,54] etc., have been upgraded/re-designed with a capability of providing unique interfacial electrochemical activity, chemical/mechanical composition information, or even unprecedented interfacial molecular structures of EDL in liquid electrolytes[53,54]. Compared to the optical and electron spectroscopy- based techniques (Raman, nuclear magnetic resonance, electron paramagnetic resonance, X-ray photoelectron spectroscopy), these liquid environment in situ AFMs already allowed to directly explore in real space multiple phenomena in otherwise obscure solid-liquid interfaces in rechargeable batteries by taking advantage of versatility of nanoprobes, with more scientific challenges waiting to be resolved by in situ/operando AFMs.

In this work, to access the whole evolution process from initial molecular-scale EDL structures, toward 3D distribution of organic and inorganic phases inside the ultrathin SEI layers, we introduce in situ electrochemical 3D nanorheology microscopy (off-resonance) combined with molecular-level force modulation spectroscopy (on-resonance). We use a matrix of two morphologically dissimilar but chemically identical surfaces of typical carbon electrode material (basal and edge graphene planes) and two solvent-electrolyte systems, polar solvent (Ethylene Carbonate, EC, mixed with Dimethyl Carbonate, DMC) and non-polar (1,4 Dioxane, DX) solvent, to get direct insight into the atomistic pictures for the underlying influence of solid-liquid interfacial nanostructure on the initial SEI formation.

## Results

### SEI formation on graphite basal and edge planes—effect of electrode surface nanoarchitecture

We choose a highly oriented pyrolytic graphite (HOPG) with an oblique polished section[55] as the graphitic anode model to observe the dynamic SEI formation on different graphite crystal planes during the lithiation. The sample preparation details can be found in Methods section and Supplementary Fig. 1. As shown in the optical image Fig. 1a, the as-polished HOPG sample has two distinct areas, the polished section, and the natural HOPG basal plane. The resulting edge-to-basal-plane ratio on the sample section is about $12 \pm 1\%$, which is several orders of magnitude higher than for the intact HOPG surface providing highly relevant model structure. The ratio of D/G Raman band on the as-prepared sample section is close to the basal plane (Fig. 1b),

indicating the highly graphitized character of the annealed section[56] (Supplementary Fig. 2 and Supplementary Note 1). In Fig. 1c, the cyclic voltammetry (CV) curves of the section sample show a typical pair of cathodic (lithium intercalation) peaks at 0.3–0.01 V and anodic (deintercalation) peak at 0.5–0.6 V, as well as an irreversible reduction peak (at 1.9–1.0 V) attributed to the SEI formation[2] which disappeared at the 2nd cycle (Fig. 1c inset). The operando Raman spectra measured on the sample section show that the G band (Fig. 1d) starts to shift at around 0.5 V during the 1st cathodic scan, confirming the onset of lithium intercalation[57]. The ion-intercalation through the HOPG edge area during the CV cycle was characterized in detail via operando Raman spectroscopy and EC-AFM (Supplementary Note 2 and Supplementary Figs. 3, 4). Figure 1e–g shows the surface topography AFM images of polished HOPG (containing section and basal plane) in the EC-based electrolyte (1 M LiPF$_6$ salt in the EC/DMC) during the first lithiation cycle. As seen in Fig. 1f, the sample section area generates multiple irregular nano-bumps with a lateral size of around several hundred nanometres, and these SEI bumps growth increases within the lithium-intercalation voltage region (0.35–0.09 V). After the CV cycles, the

HOPG edge is fully covered by electrolyte decomposition products (Fig. 1g), and the SEI thickness determined by nano-scratching on the HOPG edge is thicker than the thickness on sample basal planes (Supplementary Fig. 5a).

The high magnification AFM images in Fig. 1h further reveal the thin SEI layer formation on the sample basal planes (also see Supplementary Fig. 6 for corresponding 3D rendered images). By comparing Fig. 1h.1 and Fig. 1h.2, one can find that some irregular nano-particles form during the initial cathodic scan, accompanied by the "swelling" of graphene sheets steps. These steps continue to swell during the ion-intercalation (Fig. 1h.3–4), whereas some SEI bumps on the basal plane partially disappear during the anodic scan (Fig. 1h.5–6). This SEI removal is most likely due to the SEI re-oxidation during the anodic scan[2,46], weakening the SEI bond with the graphite basal plane (partially removed by AFM tip) and increasing its solubility in the electrolyte. A closer examination of the SEI decomposition induced carbon step swellings can be found in the section lines in Fig. 1i, j. Initially, this carbon step height was around 3 nm (-10 carbon layers), whereas the step edges' height increased by almost a factor of three (9 nm) after the

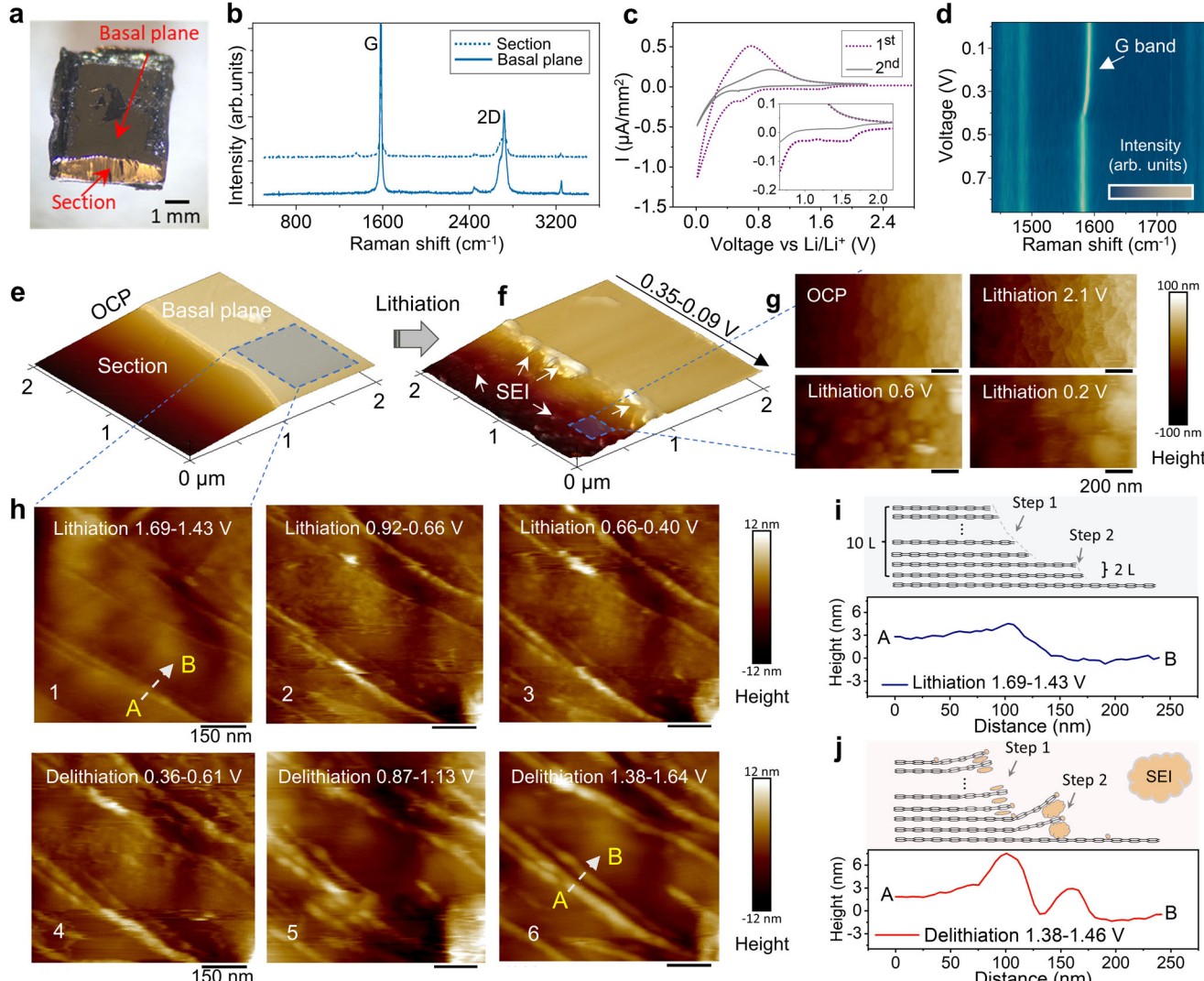

**Fig. 1 | Graphite crystal model sample and SEI formation on the edge section vs basal plane. a** Optical image of a cross-section HOPG model sample. **b** Raman spectra at polished section and basal plane of as-prepared model sample. **c** First two cyclic voltammetry (CV) curves of model sample. (0.5 mV/s scan rate, 1 M LiPF$_6$ salt in the EC/DMC = 1:1 vol% electrolyte) (**d**) G band of operando Raman spectra at section area upon the first lithium intercalation. **e**–**h** Dynamic topography

evolutions during the 1st CV scan in ethylene carbonate (EC) based electrolyte, high magnification atomic force microscopy (AFM) images of SEI formation on sample (**g**) edge planes and **h** basal planes. **i, j** Cross-section profiles and schematics of the graphite atomic step wrapping in (**h**) at different voltage ranges 1 and 6, correspondingly.

first CV scan. This can be unambiguously attributed to the decomposition of the co-intercalated solvents at the very end of graphite edge planes[2,58], since lithium intercalations can only generate a maximum of 10% interlayer expansion. This thick SEI seals the carbon edge planes by the electrolyte decomposition products with different stiffness (Supplementary Figs. 5b, 7 and Supplementary Note 3), and it is the same type of SEI formed on the sample section in Fig. 1g. In conclusion, the electrolyte is prone to decompose on the intercalation-active edge planes, especially within the solvent co-intercalation voltage range. By contrast, the graphite basal plane shows a less noticeable electrolyte decomposition.

## Subsurface 3D SEI structure revealed by electrochemical nano-rheology microscopy

The traditional operando EC-AFM characterization in Fig. 1 indicates that the SEI with differing thickness/structures are formed on the sample section (edge planes) vs terrace (basal planes). To further understand resultant SEI structures, especially how soft organic and stiff inorganic components are distributed inside the SEIs, we used in situ 3D nano-rheology microscopy (3D-NRM) to access the effective shear modulus, ($G$), and viscosity, ($\eta$), distribution underneath the SEI top surface. 3D-NRM, working in off-resonance mode, is based on the combination of shear force modulation microscopy[40,59] and force–volume measurements, in which a nanoscale tip penetrates the SEI while dithering laterally at the kHz frequency and a few nanometre amplitude (Supplementary Fig. 8). By measuring the indentation-depth dependence of the resulting in-phase and out-of-phase components of the force acting on the tip, the $G$ and $\eta$ as a function of tip apex position can be quantified. Due to very small dithering amplitude, relatively fast oscillation period (~1 ms) and slow measurement cycle (~1 s), the response of the 3D SEI has shown to be practically independent on the approach cycle frequency (varying from 0.1 to 10 Hz). At the same time, the response is clearly dependent on the dithering frequency allowing to reliably evaluate the viscous (loss) and elastic (storage) component of the SEI viscoelastic response. Each vertical probing takes data from particular area (roughly $20 \times 20$ nm², with next measurement at the safe distance of about 50 nm apart, for thinner SEI the dimensions will be correspondingly smaller). The probed area is disturbed and not probed again, while still providing a fully representative 3D snapshot of the particular state of the SEI (see detailed description in Supplementary Notes 4–6 and Supplementary Figs. 9, 10 for both methodology, typical indentation force curve, and calibration).

Figures 2a, c are the 3D-NRM measured $G$ and $\eta$ distribution along the depth of SEI layer on the sample section and basal planes, respectively. On the sample section, the overlapped $G$ vs tip-substrate distance ($Z$) spectra (Fig. 2a) show three distinct regions, representing the three different tip-SEI contact states. When the tip is in the bulk electrolyte (region I), the measured $G$ value is near zero, then increases when the tip contacts the top SEI layer at the range of around 26–100 nm from the anode surface. Between 0 and 26 nm from the surface (region III), $G$ remains uniform when the tip embeds into a stiffer ($G \approx 0.3$–$0.5$ GP) solid-like underlying SEI layer, meanwhile, the viscosity value drops to about zero as the tip reaches the stiff and mostly elastic underneath layer. This indicates a multi-layer SEI structure, which has a soft top organic layer with a larger relaxation time constant and a stiff inorganic layer with a much smaller time constant that is formed on the sample section. The relaxation time constant, $\tau$, reflects the viscosity to elasticity ratio of tip-SEI junction during the indentation at the excitation frequency with $\tau = \frac{\eta}{G}$, therefore the time constant (or shear phase signal) can be used to reconstruct the organic/inorganic SEI distribution inside the SEI layer (Supplementary Note 5). As shown in the reconstructed phase distribution image Fig. 2b, a clear two-layer structure can be found (see also Supplementary Movie 1). The larger

phase delay ($\tau > 1$ ms) SEI layer (orange colour in Fig. 2b) can be found between the electrolyte (dark green) and the low viscosity (low relaxation time constant, $\tau \leq 1$ ms) SEI layer (bright yellow). The high magnification image of the SEI shows the organic top layer as a loose and porous layer[37], different from the dense stiff inorganic underneath layer[4]. By masking out the electrolyte in the reconstruction map, the section view representation of the measured 3D data (Fig. 2b) reveals the internal structure of this multi-layer SEI that is generally laterally homogeneous within the measured area.

By contrast, a 3D map of SEI formed on the basal plane shows a structure resembling a mixed organic/inorganic monolayer SEI structure[51] across all SEI thickness within the scanning area (Fig. 2d). In Fig. 2c, the effective modulus and viscosity spectra present only two regions, corresponding to the two types of tip-SEI contact states as illustrated in the lower part of panel Fig. 2c. The average effective moduli and viscosities within this SEI layer are close to the corresponding values of the inorganic and organic layers formed on the edge planes, respectively. The measured effective moduli values agree well with the previous measurement in liquid[60,61], but are slightly smaller compared to the values measured in the post-mortem dry state[62,63], indicating some soluble organic components may be dehydrated in the post-mortem AFM analysis. The 3D reconstructed images in Fig. 2d further reveal the 'mixed' monolayer character of this SEI layer. The sectional 3D view of these SEI structures can be found in Supplementary Movie 2. The multilayer (edge plane) and monolayer (basal plane) SEI structures were also independently confirmed by AFM nano-scratching (Supplementary Fig. 7).

The SEI chemical structures were investigated by X-ray photoelectron spectroscopy (XPS) collected on the edge and basal plane surfaces (Fig. 2e–i). Relative atomic composition ratio of the SEI was determined from a fitting analysis of the *C 1s*, *O 1s*, and *F 1s* peaks shown in Fig. 2e, the results of which are shown in Fig. 2f. The analysis reveals that, both sample basal plane and section area contain substantial organic carbonates, and the sample basal plane has a higher proportion of fluorine, 11.8%, compared to the section area, 2.1%. The fluorine signal can be fitted with two peaks centred at around 685.5 and 687.8 eV (Fig. 2g–g'), corresponding to the inorganic Li-F and organic C-F, respectively. The Li-F component can stem from the hydrolysis and electrochemical decomposition of $PF_6^-$ anion[2,64]. On basal planes, the heterogeneous transfer of electrons directly to the salt anions and solvent which generates the organic/inorganic electrolyte decomposition precipitates,[65] containing a certain amount of anion decomposition products including LiF. This support the previous assumption that anion decomposition is chemical decomposition occurring at a higher voltage than EC co-intercalation and decomposition[2,66], consistent with the basal plane SEI particles formation observed between 1.6 and 0.6 V (Fig. 1h). By contrast, the SEI layer formed on sample section contains more organic species. A greater percentage of organic carbonate on the sample section is supported by the increased signal at the energy range 285–288 eV in *C 1s* (Fig. 2h–h') and 531–534 eV in *O 1s* (Fig. 2i–i'), attributed to the C–O/C = O chemical environments[41]. This can be assigned to metastable organic compounds such as ROLi and $ROCO_2Li$[67,68] located on the top layer of graphite edge plane SEI, which is most likely generated by the reduction of the co-intercalated EC molecules[69], such as lithium ethylene di-carbonate (LEDC) or lithium ethylene monocarbonate (LEMC)[3], that caused the edge expansion of carbon atomic steps in Fig. 1h. In conclusion, XPS spectra suggest that the chemical composition of the basal plane SEI consists of organic carbonate mixed with a non-negligible amount of inorganic LiF[65], while the sectional area is dominated by organic compounds[69] that are generated mostly by the reduction of the co-intercalated solvent molecules.

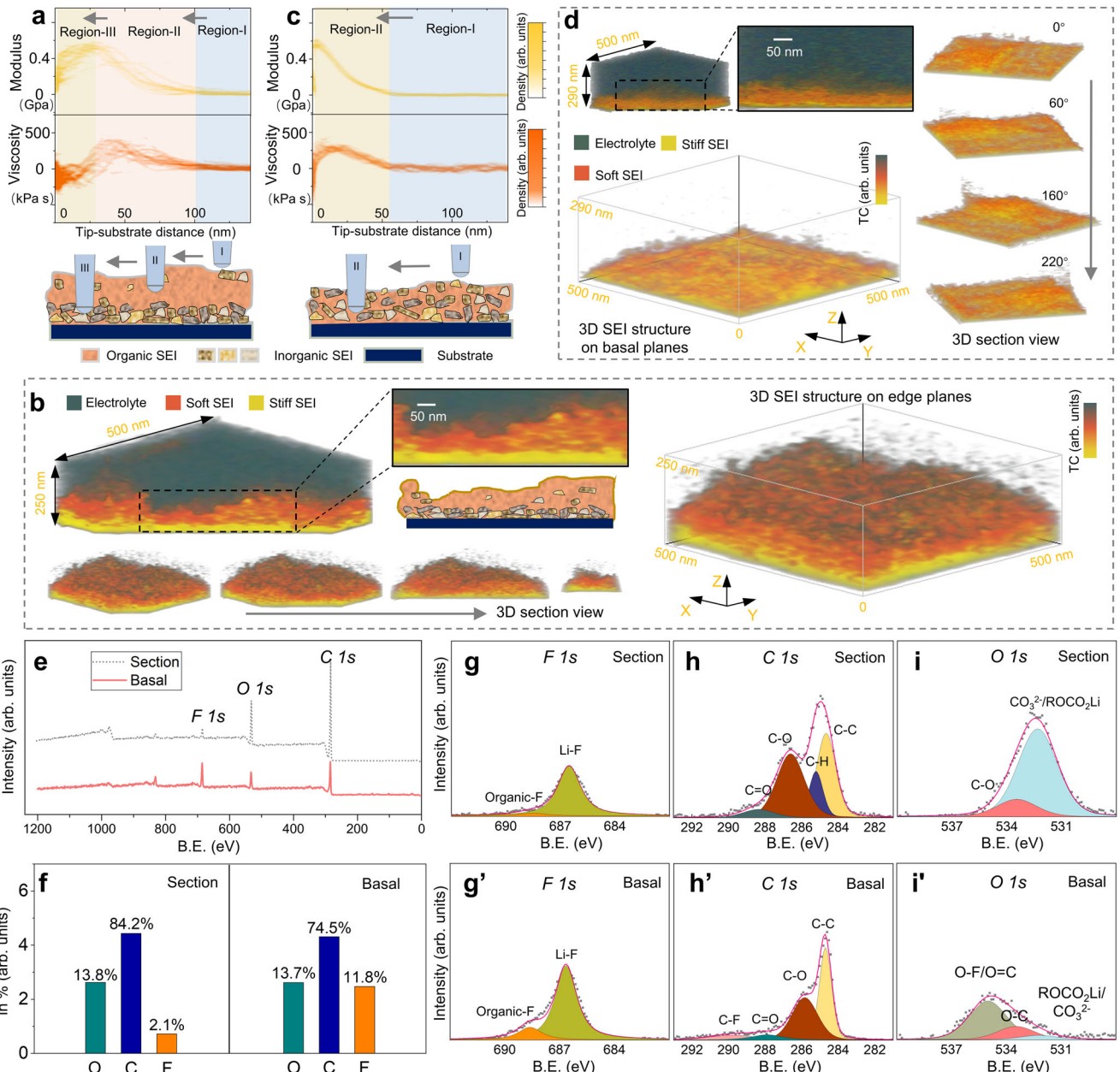

**Fig. 2 | in situ 3D nano-rheology microscopy (3D-NRM) and post-mortem X-ray photoelectron spectroscopy (XPS) characterization of SEI structure on sample section and basal planes.** 3D-NRM maps the nanomechanical viscoelastic properties of the SEI layer by penetrating it with the AFM tip. As the probing tip approaches the surface from the electrolyte, a sample is oscillated laterally at a small amplitude of few nm, and the in-phase and out-of-phase components of the lateral (shear) force acting on the tip are measured. By taking the derivative of the force over penetration increments, the effective shear modulus ($G$) and viscosity ($\eta$) distribution as a function of the depth of SEI layer is evaluated (Supplementary Fig. 8e). $G$ and $\eta$ distribution on the (**a**) edge plane section and **c** basal planes. The approximate tip SEI interaction states at each indentation depth are sketched below each panel, and the tip-sample solid−solid contact point was set as the zero-distance position. The reconstructed 3D structure of the SEI layer formed on (**b**) edge plane section and (**d**) basal planes, respectively, (the intensity of colour bar denotes the relaxation time constant (or shear phase), the green, orange and gold colours in the reconstructed 3D image represent the electrolyte, soft SEI component, and stiff SEI component, respectively). **e** XPS survey spectra located on the basal and sectional planes. **f** Elemental composition ratio obtained from survey spectrum. High-resolution XPS spectra of the *C 1s*, *O 1s*, and *F 1s* regions at the **g**–**i** section plane and (**g'**–**i'**) basal plane.

## SEI formation in weakly *vs* strongly cation solvating electrolytes −effect of EDL nanoarchitecture

In the previous section, the co-intercalated EC molecules were found to be the possible precursors of the SEI sealing the end of carbon atomic steps, resulting in distinct SEI structures on the sample section and basal planes. In other words, the graphite electrode surface nanoarchitecture (graphite crystal plane) plays the dominant role in the SEI structures in the strongly solvating EC-based electrolyte. Similarly, we also found that, this electrode surface-dependent solvent co-intercalation behaviour can happen in propylene carbonate (PC) or EC mixed with diethylene carbonate (DEC) causing carbon layer exfoliation and nanobubble formation, respectively (Supplementary Fig. 11 and Supplementary Note 7). These polar solvent co-intercalation effect results in the electrode surface structure damages and inhomogeneous SEI formed on different graphite crystal planes. To elucidate the influence of EC/PC/DEC co-intercalation on the SEI formation, we studied the SEI formation in weakly solvated electrolyte (WSE)[70] in which non-polar characteristic of 1,4-Dioxane (DX) solvents provide

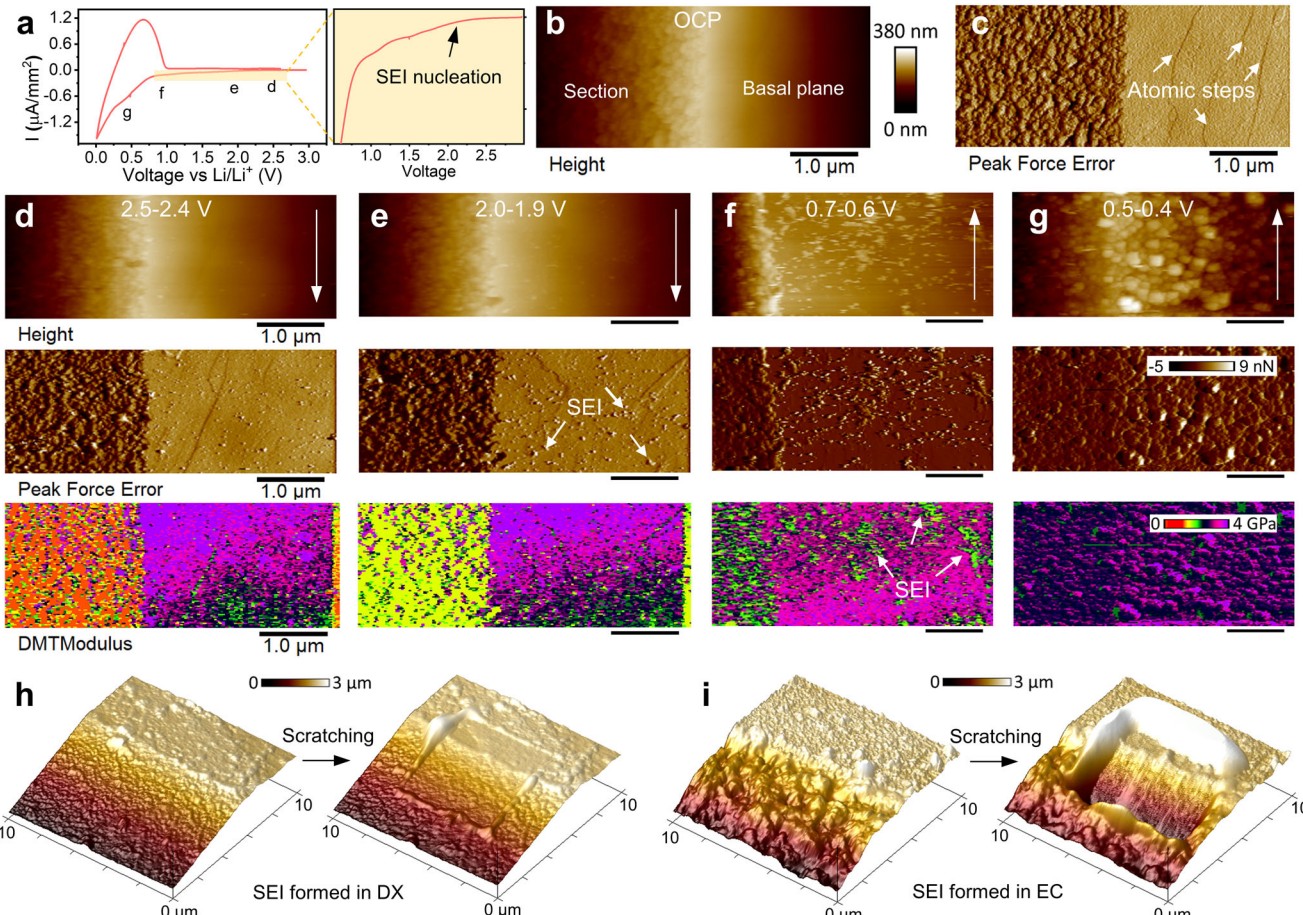

**Fig. 3 | SEI formation and mechanical property on the graphite model sample section and basal planes in the weakly solvating electrolyte (WSE) − 1 M LiFIS in 1,4-Dioxane. a** 1st CV scan curve of HOPG sample, 0.5 mV/s. **b** Topography and **c** peak force error images of the sample in the electrolyte at open circuit potential (OCP). **d**–**g** Topography, peak force error and DMT modulus on sample section and basal plane during the 1st cathodic scan in DX-based electrolytes at different voltage ranges denoted in (**a**). (Arrows in the topography images indicate the AFM tip scan direction). AFM nano-scratching of the SEI layers formed on the sample surface in **h** the DX-based vs **i** EC-based electrolyte.

small de-solvation energy for intercalations, and thereby the co-intercalation is expected to be inhibited.

Figure 3a–g show the operando EC-AFM measurements of the sample in WSE during the 1st CV scan. Figure 3b, c are the topography and peak force error (PFE)[71] images at open circuit potential (OCP). The "fish scale" like annealed carbon-steps structure can be observed on the section area, while the basal plane is relatively clean and smooth. In the voltage region 2.5 → 2.4 V (Figs. 3d) and 2.0 → 1.9 V (Fig. 3e), PFE images reveal that small nanoparticles start to form on the basal plane, meanwhile, the elastic modulus mapping indicates a soft surface adsorption layer on the sample section. Interestingly, the nano-particles on the basal plane keep growing and become visible during the 0.7–0.6 V range (Fig. 3f). These SEI bumps on the basal plane and sample section have similar modulus values of about 1.7 GPa (Supplementary Fig. 12), which indicate that it is a stiff anion-derived inorganic SEI[16,68,72]. Additionally, when further scanning down to the low voltage region (Fig. 3g), a compact SEI uniformly covers both the edge section and basal plane. This implies that the SEI formation in DX-based electrolyte is less relevant to the graphite crystal planes compared to the SEI formation in the strongly solvating EC-based electrolyte. The comparison of SEI thickness and homogeneity on the sample surface in the two electrolytes are shown in Fig. 3h, i, respectively. A thinner and more homogenous SEI layer can be found on the sample surface cycled in the DX-based electrolyte. The XPS characterizations also suggested a homogenous SEI layer in which similar

chemical components were formed on the sample section and basal plane, and the SEIs contain mainly FSI⁻ anion decomposed inorganic products, such as LiF, $Li_2S_2O_4$ and $Li_3N$ (Supplementary Notes 8, 9 and Supplementary Figs. 13, 14).

The above in situ/operando interface characterizations highlight the prevailing importance of electrolyte solvent in initial graphite SEI formation. The solvent co-intercalation induced SEI was inhibited by introducing the weakly solvating DX solvent which radically modifies the Li-ion's solvation structure. The different solvation structures in EC- and DX-based electrolyte bulk were supported by our molecular dynamics (MD) simulation and Raman spectra (Supplementary Notes 10, 11), agreeing with the proposed empirical/phenomenological interpretations that primary cation solvation sheath acts as the precursor of SEI[28,68,70,73,74]. To further develop a deeper/micromechanical understanding of cation solvation structure near the electrode surface and the resultant anion reduction process, we evaluated the EDL nanoarchitectures in the EC- and DX-based electrolyte by combining on-resonance force-distance spectroscopy and MD simulation.

Figure 4a–d show the force-separation (FS) curves[75–77] obtained on a negatively charged graphite substrate in EC- and DX-based electrolytes as well as tentative models. By approaching the tip towards the negatively charged graphite surface, the FS curves recorded the force oscillatory profile (Fig. 4a, b) during rupturing the interfacial layers. The on-resonance force curve fluctuations arise from the changes of the density/charge of molecular packing in the electrolyte near the

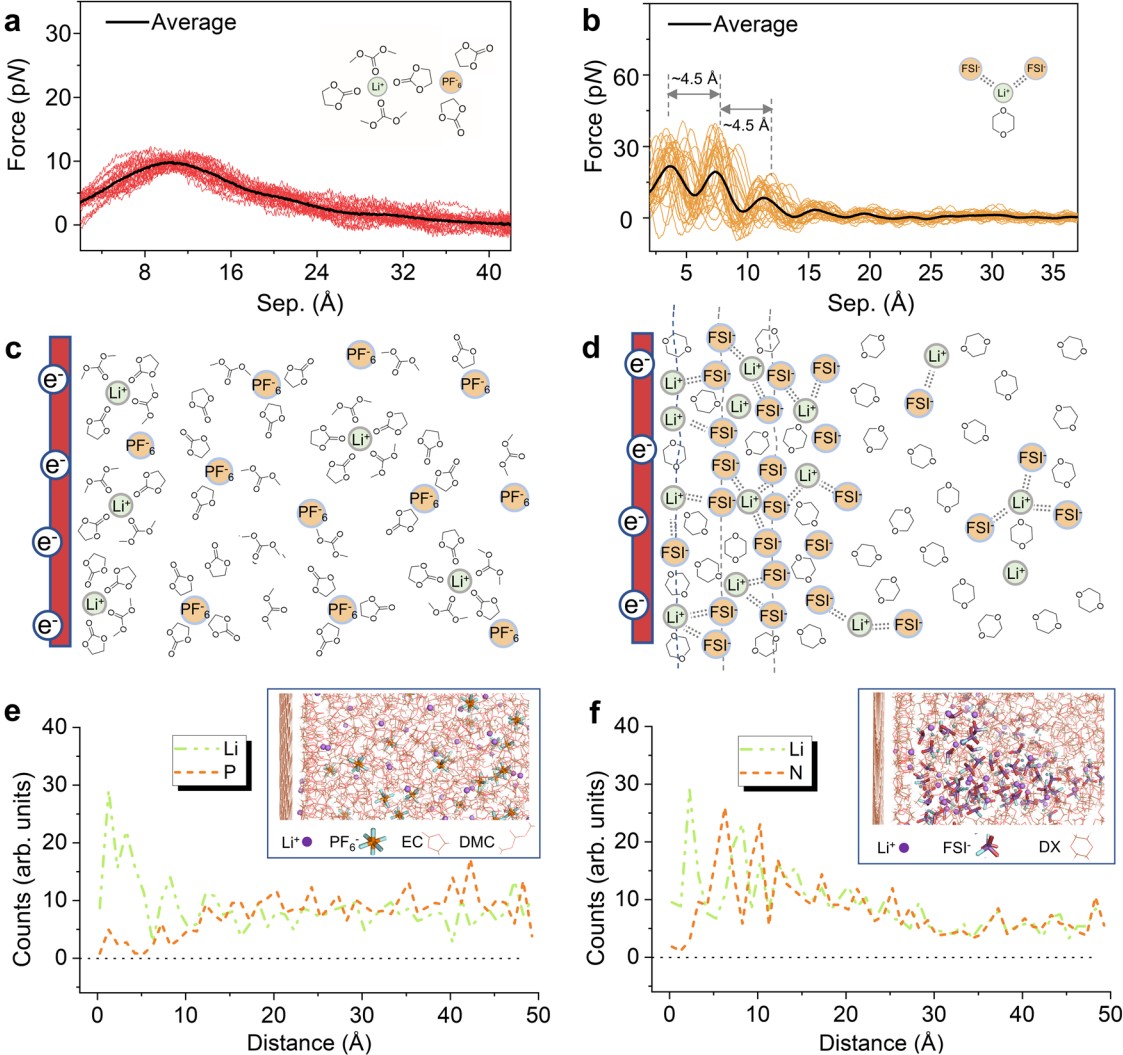

**Fig. 4 | Interfacial layering structure of EDLs through AFM force-distance spectroscopy and MD simulations.** Overlapped experimental force-separation (FS) curves associated with the perturbation of nanoscale EDL on graphite basal plane surface in (**a**, **c**) EC-based electrolyte and (**b**, **d**) DX-based electrolyte, respectively. Insets in (**a**, **b**) are the schematic EDL solvation structure according to the MD simulations. $PF_6^-$ and $FSI^-$ in (**c**, **d**) represent the hexafluorophosphate and bis(fluorosulfonyl)imide anions, respectively. Density profile of the cation and anion distribution near a carbon electrode surface in EC-based (**e**) and DX-based (**f**) electrolytes. Insets are the snapshots of the EDL structures in the MD simulations (see Supplementary Notes 10 and 11). The electrolytes are confined between two flat carbon electrodes with the left electrode negatively charged.

graphite surface, which can be used to deduce the fine nanoarchitectures within the EDL[75,78]. As shown in Fig. 4a, the EDL in the EC-based electrolyte shows that a molecule/ion packing layer generated a repulsive peak force (~10 pN), with the layer thickness being about 22–32 Å close to the solid surface. This adsorption layer can be attributed to the solvent-shared ion pair in the EC-based electrolyte. For example, as shown in Fig. 4a inset, one possible structure will be the EC/DMC molecule solvated separated ion pairs[79,80], the lithium-ion shares one of its solvated EC molecules through the $H^+$ with a $PF_6^-$ solvated by another two EC/DMC molecules[81]. The schematic EDL structure mode is depicted in Fig. 4c. In this EDL structure, the lithium ions are strongly solvated with solvent molecules, therefore, the solvated solvent molecules with reduced reductive stability might act as the precursor of the initial SEI. By contrast, as shown in Fig. 4b, the force curves in the DX-based electrolyte show visible oscillations during the rupturing of EDL, indicating a compacted and relative well-organized molecule/ion packing structure. This EDL contains the molecule/ion packing layers with the thickness of each layer being ~4.5 Å, which is close to the radius of the Li-FSI⁻ aggregate ion pair[82]. The perturbated arrangement diagram is shown in Fig. 4d. To gain further

insight into the solvation structure effects on interfacial nanoarchitectures, we compared the atomistic interface structure of the ideal models consisting of EC- and DX-based electrolytes by MD simulations. MD simulation also confirms a compact lithium-anion coordination structure near the charged electrode surface in DX-based electrolyte which provides a low dielectric constant environment. Inside this unique EDL, Lithium-ions most coordinate with two anions through bidentate coordination (Supplementary Figs. 15e, f). Moreover, Fig. 4e, f shows the ion number density profiles of cations ($Li^+$) and anions near the negatively charged carbon electrode surface (generated through the fluorine or nitrogen atoms). For the EC-based electrolyte, the number density profiles show that the Li-ions and solvent molecules aggregate on the inner layers of EDL, while the anions are squeezed out from these layers. However, as shown in Fig. 4f, a significant number of nitrogen ($FSI^-$) in the inner EDL layer can be found in the DX-based electrolyte. The periodicity of the cation and anion density profiles (~4–5 Å) are consistent with the FS oscillation in Fig. 4b, indicating that an ordered structure was formed on the DX-based electrolyte EDL. It is also worth noting that, comparing the radial distribution function (RDF) in the near surface region (EDL) and electrolyte bulk, the

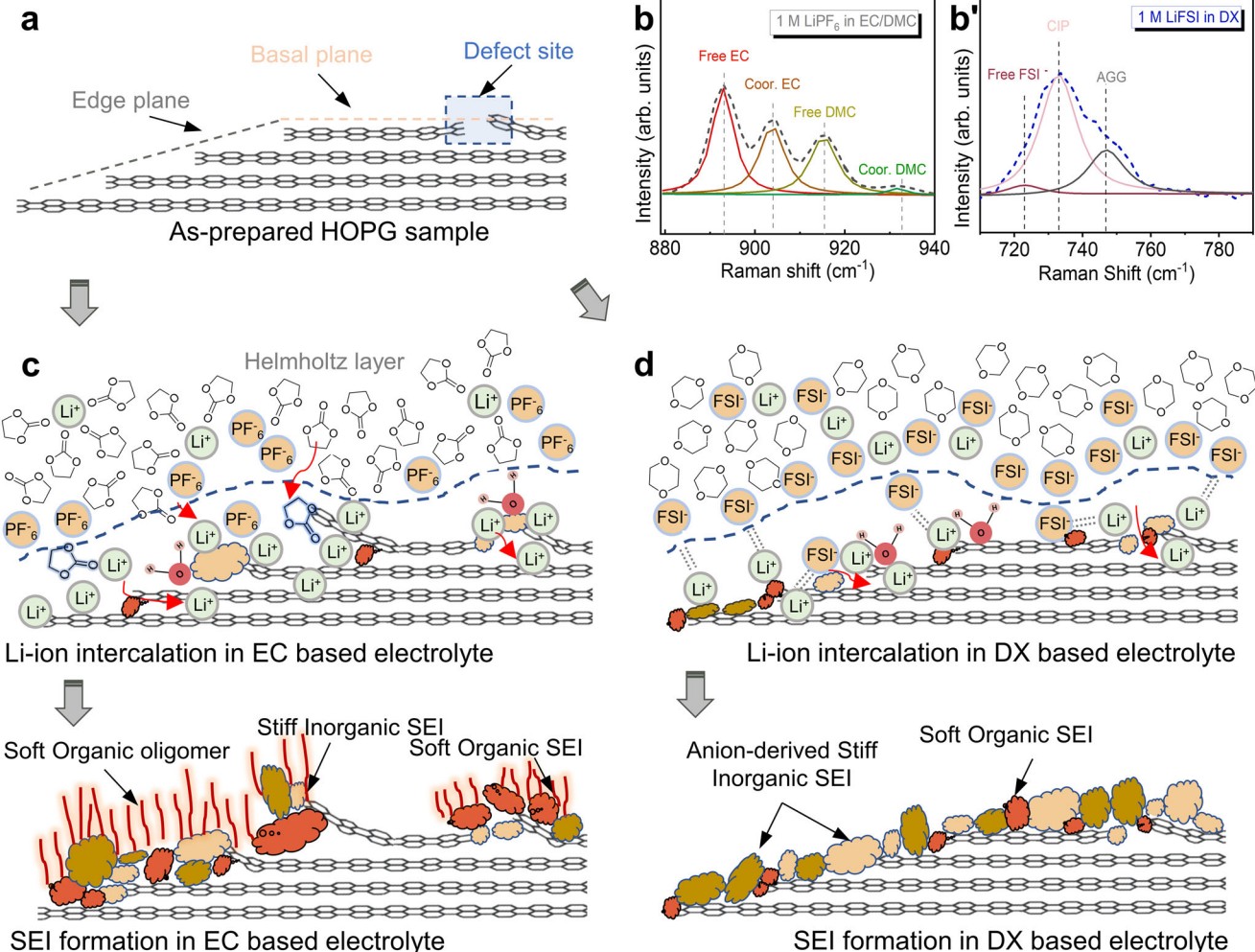

**Fig. 5 | Schematic diagram representing the nanoscale and atomistic picture of initial SEI formation mechanisms. a** As-prepared HOPG model sample with calibrated graphite edge plane and basal plane (scattered natural defects may exist on the graphite basal planes). Raman spectroscopy characterization of two electrolyte solvation structures in bulk, **b** C–O stretching vibrational band of EC-based electrolyte and (**b**′) S–N–S stretch band of DX-based electrolyte. CIP and AGG represent contact ion pair and aggregate, respectively. Models of initial SEI formation mechanism on model samples matching the study results, **c** strongly solvating EC-based electrolyte and **d** weakly solvating (DX-based) electrolyte (WSE). Dimethyl carbonate (DMC) molecules are not shown in the sketches.

coordination number (CN) of $PF_6^-$ anion with Lithium-ion drops to nearly zero in EC-based electrolyte inside the EDL (Supplementary Fig. 15a, b), while the CN of $FSI^-$ remains higher than DX solvent (Supplementary Fig. 15c, d). This confirms that the solvent molecules still dominate the first solvation sheath within the EDL in EC-based electrolyte and anions are repulsed further away from the negatively charged electrode surface[83,84], but due to the weak coordination effect between DX and lithium (relatively stronger coordination effect between $FSI^-$ and lithium), $FSI^-$ still dominates the first solvation sheath inside the EDL.

## Discussion
### Nanoscale and atomistic picture of initial graphite SEI formation
By cross-examining the effects of the electrode and solvation structures on SEI formations, we now can summarize the role of nanoscale electrode features, solvation structures, and EDL nanoarchitecture factors of the SEI formation process in the graphite model sample, as well as the new insights that can be extended to various electrolyte/electrode interfaces using AFM dynamic force modulation spectroscopy.

As shown in Fig. 5a, the as-prepared HOPG model sample has an 'intercalation-active' oblique section with plane edges and an intercalation-inactive basal plane. Li-ion ions can only intercalate into

graphite interlayers through the edges (or some defect sites on the basal planes). Meanwhile, according to the Raman spectra in Fig. 5b, b′ and detailed discussion in Supplementary Note 11, the Li-ions in the commercial EC-based electrolyte are well-solvated by solvent molecules (mostly EC, see Supplementary Fig. 16) but mainly coordinate with anions to form contact ion pairs in the DX-based electrolyte[70] (Supplementary Fig. 17). As a result, in the EC-based electrolyte, the SEI formation is highly crystal plane dependent due to the solvent co-intercalation effects. The graphite basal plane (no intercalation) SEI is mainly derived from the precipitation/adsorption of electrolyte decomposition products at the defect sites or the dangling bonds, passivating the surface's chemically active sites. This SEI can form simultaneously once the fresh carbon surface is exposed to the electrolyte (see Supplementary Movie 3) due to the mismatch of the electrolyte's lowest unoccupied molecular orbital (LUMO) energy level and anode surface potential. On the intercalation-active edge section, the strongly solvating solvent molecules partially co-intercalate with Li-ions into the graphite interlayers, which not only causes the warping of carbon layer edges (Fig. 1h–j), but also solvent decomposition as LEDC/LEMC that can further decompose into inorganic species, such as $LiOH/Li_2O/Li_2CO_3$[2,3] (Fig. 5c). The preferential decomposition of solvent molecules facilitates the formation of the soft organic oligomer outer layer[4] on the graphite edge

planes. This solvent co-intercalation dominated SEI formation sequence results in a layered structure on the intercalation active edge plane areas, with a viscoelastic organic top layer (viscosity $\mu \approx$ 100–300 kPa·s) and inorganic underneath layer (effective shear modulus $G \approx 0.3$–0.5 GPa).

In the DX-based electrolyte, as shown in Fig. 5d, since the co-intercalation effect is inhibited, the SEI formation processes are dominated by the nanoarchitecture of EDL, rather than the crystal plane-dependent solvent co-intercalation. The dominating coordination role of $FSI^-$ anion, rather than DX solvent, is preserved on the negatively charged electrode surface, despite a decreased anion CN within EDL (Supplementary Figs. 15c, d). During the electrode polarization, lithium ions carry the $FSI^-$ solvation sheath moving toward the electrode surface, and the coordinated $FSI^-$ accumulates and builds up the negative charge layer near the electrode surface. The $Li^+$-$FSI^-$ contact ion pairs and aggregates packing near the electrode surface facilitate the anion reduction on both basal and edge planes before the electrode potential dropping to solvent decomposition. As a result, the SEI formed on the edge section and basal planes are relatively identical: a thin anion-derived inorganic-rich SEI layer with the higher $G \approx 2$–4 GPa. Moreover, comparing the SEI formation in the EC-based solvent with different LiFSI concentrations (Supplementary Fig. 18), we, therefore, suggested that it is the solvation structure (competitive coordination effects of solvents and anions with Lithium-ions), rather than the anion species, that dominates the preferential electrolyte decomposition process in these electrolyte systems.

## Insights on SEI formation mechanisms at various electrode/electrolyte interfaces via force spectroscopy

The stimulating feature of the current study is merging the study of SEI structure, electrode surface structure and initial EDL structure using complementary combination of two dynamic force microscopies/spectroscopies. This direct link of the EDL structure and three-dimensional nanoscale SEI structure was possible due to 3D-NRM−dynamic modulation force spectroscopy with 3D spectroscopic resolution[53]. We shall stress that there are two essential capabilities required for studying these two important interfacial structures. First, for the molecular level EDL structure, one can use traditional static force−distance curves used in AFM that characterize the n$N$ range tip-EDL interaction forces in either highly concentrated electrolyte or ionic liquid electrolytes[24,85–87] on the charged electrode. Things get complicated by the fact that in commercial dilute (1 M) electrolytes and small organic molecules as solvents, the molecules/ion packaging layers have a much smaller interaction force at around p$N$ range, requiring frequency/amplitude modulation mode spectroscopy[75,76], and a high-quality factor cantilever (or tuning fork) in the EC-AFM for accessing these EDL structures. Secondly, the fine three-dimensional structure of SEI requires a 3D nanomechanical force spectroscopy that is sensitive to the wide range of mechanical properties (from <MPa to several GPa). The Peak Force Tapping force spectroscopy only explores the surface SEI Young's modulus[88], while the subsurface SEI structure can be detected by the approach-retract force−distance spectroscopy in the elastic deformation region using one-step or two-step indentation mode[35,37,38]. In this case, it is the inorganic component that is typically detected[33], but the soft organic SEI component (especially the gel-like part of SEI[41]) generally un-detected, morphologically not resolved and mechanical properties unknown.

The capability of resolving the organic and inorganic part of SEI becomes increasingly significant facilitating the key idea for designing advanced electrolytes, mentioned in the introduction (SIS/WIS/LHCE/ULCE). It is, in general, similar to the weakly solvating DX-based electrolyte, which aims at forming anion-derived inorganic SEI by increasing fluorine rich anion occurrence inside the first solvation sheath, and therefore the reductive stability of solvents inside the EDL

can be enhanced[18]. This anion-derived SEI formation is confirmed, by operando EC-AFM/force-spectroscopy, not only happening on the plating/stripping-based Li-metal anode (Supplementary Note 12 and Supplementary Fig. 19) but also on different graphite crystal planes, indicating the universal characteristic of anion preferential decomposition as an electrode-electrolyte interface design and optimization rule.

Contradictory, it is also worth mentioning that the solvent decomposition-derived organic SEI can also be a good interfacial passivation layer. For example, the ultra-low concentration electrolyte (ULCE) was recently suggested to be a promising low-temperature electrolyte[17] because of its low de-solvation energy and preferential decomposition of solvents which results in the formation of porous oligomers and elastic alkyl carbonates that have higher cation conductivity than inorganic species. This preferential decomposition of solvent in ULCE shares the same electrolyte decomposition paths as the strongly solvating EC-based electrolyte. In this work, according to XPS and 3D-NRM results in Fig. 2, we believe that the rich oxygen functionalized organic SEI layer (on graphite edge planes) with porous structure may be essential for minimizing the de-solvation/re-solvation energy barriers in the strongly solvating electrolyte, especially when the de-solvation becomes a significant obstacle in sub-freezing temperature[89]. In essence, graphite edge plane is energetically favourable to forming a porous and elastic alkyl carbonates organic SEI outer layer playing a major role in mitigating the abrupt cation solvation structure changes between the electrolyte and graphite lattice.

From this point, the trade-off between structural mechanical stability, ionic conductivity and de-solvation/re-solvation capability of SEI layer must be considered simultaneously when optimizing an electrode-electrolyte interface. Although it is still too early to conclude whether anion-derived inorganic SEI or solvent-derived organic SEI component is responsible for the fast ion transportation inside SEI and which component (or what kind of organic/inorganic mixed structure) is beneficial for battery cycle and rate performance, our rheology microscopy with organic/inorganic resolution can be a powerful and universal platform for further exploring the mechanisms underlined the electrolyte solvation (or electrode surface) structure guided SEI formation and battery performance.

This work opens a promising area for the direct study and observation of reactive solid-liquid interface using AFM with 3D nano-mechanical properties probing. There are still many conjectures and assumptions about the alkali metal solvation structure in the EDL and the resulting SEI formation mechanism, which are significant for battery performance, but still in the early stage of theoretical model establishment. For example, the cation solvation complexes structure differences in bulk electrolyte and EDL, the kinetics of de-solvation processes under the polarized electric field, the reductive decomposition of the molecules/ions of the EDL adsorbed layer to form the SEI film, etc., which are all in the hardcore of the alkali rechargeable battery system. Quantitatively characterizing these physical and chemical processes experimentally and establishing appropriate physical models to describe them are frontier scientific challenges that require more effort in future battery research. Further incorporating these dynamic force modulation spectroscopy with a Raman/IR/THz optical spectroscopy by a light-confined nano-probe[51], or electrochemical spectroscopy by a micro-pipet probe[48,49], can be a direction for extending the chemical structure understandings of these interfaces, fulfilling the mechanism underlined the solvation structure and electrode surface structure guided interfacial electrochemical reactions. This can further provide the optimization guidance for various new solid-liquid interfaces, such as electrochemical $CO_2$ reductive catalysis, corrosion, lubricant, supercapacitor, small molecule transport in bio-meniscus, etc.

## Methods

### Sample preparation

HOPG samples were purchased from SPI (438HP-AB, SPI-1 Grade). Supplementary Figure 1a shows the diagram of sample preparation and characterization processes. An artificial micro-section was created on the HOPG sample by polishing one side of the HOPG block using Leica TIC-3X (Leica, Germany) cross-section polisher. The section angle is adjustable by tilting the sample stage. As shown in Supplementary Fig. 1a, the as-polished HOPG has two distinct areas, which are the sample section area (consisting of controlled ratio of graphite edge to basal planes) and sample terrace area (consisting of predominantly graphite basal planes). After polishing, the HOPG sample terrace area was then cleaned by using Scotch tape to remove the impurity deposition layer. To obtain an electrical and ionic conductive section, the as-polished samples were annealed at 500 °C for 1 h in Air and Argon to remove the amorphous carbon layer[90]. In this study, the obtained section angle is ≈ 6–7 degrees.

### Electrochemical environment SPM, Raman, and XPS characterizations

After the BEXP polishing, the as-polished HOPG sample, with a thickness around 0.5 mm, is horizontally mounted on the custom-made AFM electrochemical-cell in an AFM Multimode system, (Bruker, USA) as shown in Supplementary Fig. 1b. The manufacturing drawings of the AFM electrochemical-cell are provided in the public repository (see the data availability statement). Cell open circuit potential was recorded for 300 seconds after the injection of electrolyte (-0.2 mL), and then about 200 nN force was applied to the tip to clean the electrode surface by scanning in a $15 \times 15\ \mu m^2$ area before the operando EC-AFM measurements. Reference 600+ potentiostat (Gamry, USA) was used for cyclic voltammetry (CV) and electrochemical impedance spectroscopy (EIS) measurements. CV data were obtained at a scanning rate of 0.5–0.7 mVs$^{-1}$ between OCP-0.01 V vs Li$^+$/Li, using polished HOPG as a working electrode and lithium metal as reference and counter electrode. 1 M LiPF$_6$ in ethylene carbonate and dimethyl carbonate (EC:DMC) = 1:1 (v:v) was used as the regular electrolyte, 1 M LiFSI in 1,4-dioxane was used as a weakly solvated electrolyte. The electrolytes using LiFSI salt disolved in EC/DMC solvents with different concentrations of 0.1 M, 0.5 M, 1.0 M, 1.5 M, 3.0 M, 6.0 M are prepared to study the concentration dependent solvation structures and SEI formation (see Supplementary Fig. 18a). The SEM images were captured by an SEM JSM-7800F (JEOL, Japan). XPS was performed using a Kratos Analytical AXIS Supra spectrometer with monochromatic Al $K\alpha$ 1486.7 eV X-ray source, operating at 15 kV, 15 mA, and equipped with an electron gun for charge neutralization. Depth profiling measurements were performed using an in situ Ar Gas Cluster Ion Source (GCIS; Kratos Analytical Inc. Minibeam 6) operated at a cluster size of Ar1000+ with impact energy of 10 keV, equating to an energy of 10 eV per atom. For the ion beam, a raster size of $3 \times 3$ mm$^2$ was employed. All the spectra were analyzed using CASAXPS (Casa Software Ltd, UK). Operando electrochemical SPM was performed inside the glovebox (MBraun, Germany) with oxygen and moisture content <0.5 ppm. A custom-made electrochemical cell with a two-electrode system was used for the electrochemical measurements (Supplementary Fig. 1c). Nano-mechanical properties were measured by ultrasonic force microscopy and peak force quantitative nanomechanics (QNM) in a liquid environment. 3D-NRM was carried out with a real-time data acquisition board and LabView software (National Instruments, USA). The tips used were Bruker ScanAssyst-fluid$^+$ and force modulation tips (Au coating) with a force constant of 0.5 and 3–40 N/m, respectively. The amplitude modulation force spectra of electrical double layer structures were detected using ARROW-UHF AuD-10 (Nano-World, Germany). The reconstruction and calibration methods of the FS curves can be found in Supplementary Note 13 and Supplementary Fig. 20. The bulk state solvation structures of different electrolytes are measured by Raman spectrometer (LabRAM HR Evolution). Operando Raman spectra during the ion-intercalation were measured using a green laser with 532 nm wavelength using an EL-CELL (ECC-Opto-Std, Germany). A PDMS reference sample (sylgard-184) was choose as the 3D-NRM calibration sample. Details of 3D-NRM operation and calibration are in Supplementary Notes 4–6.

### Finite elements modelling and molecular dynamic simulation

COMSOL Multiphysics® software with MEMS and Structural Mechanics module was used to model the 3D-NRM calibration data. In the COMSOL simulation, the deformation of the sample was prescribed at the fixed oscillation frequency, whereas the forces acting on the stationary tip were integrated. The simulation results of the displacement of the tip-PDMS junction at different tip indentation depth can be found in the Supplementary Movie 4. MD simulations of EC- and DX-based electrolytes were performed using GROMASC software. The temperature was controlled at 300 K using the Nose-Hoover thermostat[91]. The many-body force field OPLS-AA[92] was used. The partial atomic charges of PF$_6^-$, FSI$^-$, DX, EC/DMC, are obtained from refs. [93–96]. The electrostatic interactions were computed using Particle-mech Ewald (PME) methods. The cut-off distance of 1.5 nm was adopted for electrostatics and Van der Waals interactions. In EC-based electrolyte, the negative charge ($\Delta Q = 0.0036$ e) is added uniformly to each carbon atom in one of the electrodes to generate the polarization field. The dielectric constant of EC- and DX-based electrolytes are around -25[97] and -2.2[70], respectively. Therefore, to simulate a similar voltage bias at the order of a few voltages on the carbon electrodes, we $\Delta Q = 0.00032$ e negative charges to each carbon on the electrode for DX-based electrolyte system. Our electrode polarization voltages (a few voltages) for EC-based electrolyte were estimated using the charge density provided elsewhere[98], while a more precise method for calculating the electrode voltage can be found in the corresponding reference[99]. Note that, too large polarization potential/charge may result in the depletion of Li-ions and anions in the bulk region, in which much less ions can be found, compared to systems with uncharged (or weakly charged) electrodes. The simulation boxes were $5.145 \times 5.105 \times 15$ nm$^3$ with 10 nm the dimensions between the two electrodes (see Supplementary Figs. 21a, b). A 10 nm distance between two electrodes was chosen such that the forces on the molecules in the middle of the simulation box were statistically the same as those in the bulk simulations without electrodes. In every simulation result, we also examined the solvation structures of lithium-ions in the polarized bulk electrolyte by radial distribution function (RDF), which is consistent with that of bulk electrolyte under the unpolarized state (see Supplementary Figs. 16, 17). Namely, the half distance of the two electrodes was controlled to be larger than the Debye length of EDL (around 2–3 nm according to the force curve experiments). Besides, the electrode area size was chosen to ensure the statistically continuous in-plane periodic structure of the carbon hexagonal rings in two electrodes. The asymmetry direction, defined as the axis perpendicular to electrode surfaces, will be referred as the Z axis. The simulation boxes were periodic along the X and Y directions. Each electrode was represented by double-layer carbon electrode with the (001) basal face exposed toward the electrolyte. In the simulation boxes, the EC-based electrolyte consists of 153 LiPF$_6$, 1187 EC, and 934 DMC, the DX-based electrolyte consist of 156 LiFSI and 1638 DX, corresponding to the molar concentration and solvent ratio in the two electrolytes at ambient condition (see Supplementary Note 14 and Supplementary Table 1). The equilibrium states of the system were checked by the monitoring the electrolyte density during the NPT equilibrium simulation (Supplementary Fig. 21c–f). In the initial structure, the cations, anions and solvent molecules are randomly placed between the two fixed electrodes using Packmol code. After equilibrium, a 30 ns NVT run was performed for each electrolyte with a simulation time step of 1 fs. Since positions of our electrode atoms are constrained, there are

no interelectrode relaxation/motion that can influence the local electric field. Typical snapshots of the EDL structures in two electrolytes after the MD simulations can be found in Supplementary Fig. 22. The EDL molecule/ion density profiles were calculated by averaging the five equilibrium simulation structures. The EDL solvation structure are the statistic results between $0 < Z < 2$ nm according to the measured surface force–distance spectra.

## Data availability

All data generated in this study, including descriptions of the data sets, manufacturing drawings of EC-AFM cell, have been deposited in the figshare database without access code [https://figshare.com/articles/journal_contribution/e-Data-V3/21992906]. The video data generated in this study are provided in the Supplementary Information file. Additional supporting data are available upon reasonable requests to the corresponding author.

## Code availability

LabVIEW data capture script and the developed code for reconstructing 3D-NRM force curves into tomography slices are available in the figshare database without access code [https://figshare.com/articles/journal_contribution/e-Data-V3/21992906].

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

## Acknowledgements

The authors wish to acknowledge the financial support by the Faraday Institution (grant number FIRG018, to NT, RP, and OK), EU Graphene Flagship Core 3 project and EPSRC project EP/V00767X/1 (to OK). We are also grateful to Bruker UK, Leica Instruments, LMA Ltd and John Booth from Gamry instruments for the in-depth application support of the relevant instrumentation. The authors also acknowledge scientific insights by Andrey Khlobystov, software coding support by Asher Jenners and Weijian Zhang, and NEXGENNA consortium for the new methodology development.

## Author contributions

Y.C., N.T., and O.K. developed overall methodology and carried out the data analysis. Y.C., S.G., and C.W. performed the AFM measurements. L.F., S.J., A.D., M.I., and R.P. performed the XPS studies and analysis. Y.C., F.W., and R.Y. performed the Raman studies. Y.C., M.N., and O.K. designed and performed the electrochemical measurements. W.W. and Y.C. performed the computational studies. All authors contributed to the manuscript preparation and revision. O.K. supervised the work team.

## Competing interests

The authors declare no competing interests.
