## [Peer Review File · Nature Communications]

Peer review comments, first round –

Reviewer #1 (Remarks to the Author):

The manuscript by Chen and Kolosov et.al. reports on a scanning probe-based investigation on the SEI structure, mechanical properties (modulus and viscosity), as well as solvation structure. The important information, is then the link between the solvation structure and the SEI structure, i.e., the possible SEI forming mechanism. This is an highly interesting work, and is recommended to be considered published in Nature communication once the following concerns are addressed:

(1) the so-called 3D nano-rheology microscopy is an intriguing technique, but it is invasive. With the nano probe diving into the SEI layer by hundreds of nanometers, the components in SEI is inevitably pushed around and severely disturbed. Furthermore, the measured mechanical properties, especially the viscosity, could be sensitively dependent on the loading speed of the normal pressure on the tip, as well as the shear force frequency and amplitude. It would be necessary for the authors to comments on these effects, especially in the context of this study, what effects do these factors have regarding the spatial resolution in- and out-of-plane, and how quantitatively reliable the mechanical property measurements are.

(2) the authors compared the solvation structure of EC:DMC and DX based electrolytes by observing the differences in Force Curves on electrode|electrolyte interface. This piece of information is very important in figuring out the SEI formation mechanism, but is totally downplayed in the title and the abstract of the paper. For one thing, the 3D-rheology microscopy does not deserve the current attention in the title.

(3) it will be highly interesting to use the methods reported here to study the difference between EC and PC in their interaction with graphite anode and the SEI formation. It is probably much more relevant to the battery field than DX on graphite. The authors are recommended to provide data and discussion on PC-based electrolytes.

Reviewer #2 (Remarks to the Author):

SEI is a critical but complex topic in battery research, and it is still challenging to fully understand through both computational and experimental investigation. Developing any new methodology, therefore, is very important to the field. This work presents a new method using both electrochemical atomic force microscopy and 3D nano-rheology microscopy to identify the chemical composition of SEI and illustrate its early formation process. However, I feel that some experiments are not explained very clearly, and the MD simulation part doesn't provide much useful support or insight. Here are some detailed questions for the authors to consider:

(1) On Page 3, please add which electrolyte systems to study in this work before the RESULTS section.

(2) Fig 1K, what is the peak near 160 nm?

(3) Page 6, Please add a clear discussion to explain the distinct SEI structures between the edge and basal planes via XPS studies, since this was mentioned at the beginning of the next part.

(4) Overall, I feel the figures are all too small. Some letters and features are hard to read.

(5) Figure 4c is kind of misleading. The authors always mention the EC-Li coordination for EC-based electrolyte, which is also demonstrated in Figure 4c that only the EC coordination appears in the inner-most layers next to the anode surface, it should be noted that the real electrolyte used here is 1 M LiPF₆ in EC/DMC (please also explain the molar or volume ratio between EC/DMC on page 4). It is expected that DMC also coordinates to Li, probably even at a larger number depending on the EC: DMC ratio, because the literature (Fundamental Research 1 (2021) 393-398) suggests a stronger Li-DMC binding energy than Li-EC. This should be able to validate through MD simulations.

(6) The XPS in Figure S11 e and h also shows the Li-F SEI component on both basal and sectional planes, is this due to the decomposition of anion? The relative Li-F peak is higher on the basal

plane, why?

(7) Why the similar Figures in Figure S11 are not measured for the DX-based electrolyte?

(8) Are insets figures in figures 4e and 4f from MD simulations? Li should also be highlighted.

(9) It is mentioned on page 11 that "The DX solvent molecules rarely exist inside the inner Helmholtz layer" which is not supported by the inset in Fig 4f.

(10) MD simulation methods are not described clearly which is not enough for others to reproduce the work. The MD results are also not reported and explained at all, not sure what is the purpose of the MD part. The MD results should provide detailed and clear information on the interfacial structure of the electrolyte. If the results were not discussed, this part should be removed.

We are grateful to the Reviewers for the in-depth comments and for the overall positive evaluation of the submitted manuscript. We carefully revised the manuscript and the SI full in response to the comments, with the point-by-point answers listed below. In particular, we feel that the manuscript now resolves the concerns regarding the invasiveness of the microscopy technique, interpretations and data presentation of the MD simulations, and includes more detailed discussions, thanks for the thoughtful and in-depth questions and comments. We also feel that the revised manuscript now even more clearly brings together concepts of direct observation of anode-electrolyte interface structure, from the initial atomic-scale EDL surface towards the resultant 3D nanostructures of SEI, to provide a better impact on a broader scientific community.

We highlighted all revisions in the “highlighted” version of the revised manuscript.

Reviewer #1 (Remarks to the Author):

The manuscript by Chen and Kolosov et.al. reports on a scanning probe-based investigation on the SEI structure, mechanical properties (modulus and viscosity), as well as solvation structure. The important information, is then the link between the solvation structure and the SEI structure, i.e., the possible SEI forming mechanism. This is an highly interesting work, and is recommended to be considered published in Nature communication once the following concerns are addressed:

Response REV1: We sincerely thank Reviewer 1 for the positive comments on our work, the manuscript has been carefully amended according to the Reviewer’s constructive suggestions, the point-by-point response is attached below.

(1) the so-called 3D nano-rheology microscopy is an intriguing technique, but it is invasive. With the nano probe diving into the SEI layer by hundreds of nanometers, the components in SEI is inevitably pushed around and severely disturbed. Furthermore, the measured mechanical properties, especially the viscosity, could be sensitively dependent on the loading speed of the normal pressure on the tip, as well as the shear force frequency and amplitude. It would be necessary for the authors to comments on these effects, especially in the context of this study, what effects do these factors have regarding the spatial resolution in- and out-of-plane, and how quantitatively reliable the mechanical property measurements are.

REV1_R1: We gratefully acknowledge this useful comment on invasiveness. Indeed, 3D nano-rheology is an invasive nano-indentation method, while its invasiveness is the key to accessing the subsurface structure of ultrathin SEI layers, similar to the 3D tomography AFM¹ and XPS etch milling methodology. In other words, 3D nano-rheology is a complementary methodology for the traditional electrochemical AFM^{2,3,4} which is limited to merely studying the surface topography/properties of SEI layer. In this work, we used the traditional non-invasive *operando* AFM to observe the dynamic SEI formation, and after the SEI film is grown to the particular cycle point on the electrode surface. We then performed *in-situ* (in electrolyte environment) 3D nano-rheology on the as-yet undisturbed SEI, in which each image pixel is “fresh” with undamaged SEI nanostructures. Therefore, by taking advantage of invasive nano-indentation with shear modulation and high-lateral resolution of AFM technique, 3D nano-rheology can differentiate the elastic and viscous component distribution inside the SEI layers, which is not available in traditional statistic AFM force spectroscopy techniques^{5,6,7,8}.

We thank the Reviewer’s and Editors for suggestion to describe the 3D-NRM for the broader audience, and to clarify

effects of the tip loading/modulation speed and mechanical property quantification and add the following in the main manuscript:

Page 6, lines 3-7, “3D-NRM maps the nanomechanical viscoelastic properties of the SEI layer by penetrating it with the AFM tip. As the probing tip approaches the surface from the electrolyte, a sample is oscillated laterally at a small amplitude of few nm, and the in-phase and out-of-phase components of the lateral (shear) force acting on the tip is measured. By taking the derivative of the force over penetration increments, the effective shear modulus (G) and viscosity (η) distribution as a function of the depth of SEI layer is evaluated (SI, Fig. S2.1e).”

Page 7, lines 5-9, starting with “3D-NRM, working at off-resonance mode, ...”

And the following paragraph

Page 7, lines 10-17, “Due to very small dithering amplitude, relatively fast oscillation period (~1 ms) and slow measurement cycle (~1 s), the response of the 3D SEI has shown to be practically independent on the approach cycle frequency (varying from 0.1 to 10 Hz), while clearly dependent on the dithering frequency allowing to reliably evaluate the viscous (loss) and elastic (storage) component of the SEI viscoelastic response. Each vertical probing takes data particular area (roughly 20x20 nm², with next measurement at the “safe” distance of about 50 nm apart, for thinner SEI the dimensions will be correspondingly smaller). The probed area is disturbed and not probed again, while still providing a fully representative 3D snapshot of the particular state of the SEI (see detailed description in SI sections S1-S3 for both methodology, typical experimental results, and calibration).”

We also add a detailed description of the dynamic behaviour of the 3D-NRM in the revised SI

Page S7, lines 28-51, starting with “The effect of vertical loading ...”

Page S8, Figure S3.2 with the caption.

(2) the authors compared the solvation structure of EC:DMC and DX based electrolytes by observing the differences in Force Curves on electrode|electrolyte interface. This piece of information is very important in figuring out the SEI formation mechanism, but is totally downplayed in the title and the abstract of the paper. For one thing, the 3D-rheology microscopy does not deserve the current attention in the title.

REV1_R2: We are grateful for the good suggestion. We amended the title that now reads as “Nanoarchitecture factors of solid – electrolyte interphase formation via 3D nano-rheology microscopy and surface force-distance spectroscopy” and added the following sentence to the abstract

Page 1, lines 22-25, “By probing solvent molecules - ions arrangement within the EDL and quantifying 3D mechanical property distribution of organic and inorganic components in the as formed SEI layer, we reveal the nanoarchitecture factors and atomistic picture of initial SEI formation on graphite anode in strongly and weakly solvating electrolytes.”

to highlight the links between the observed solvation structures and SEI formation mechanism.

(3) it will be highly interesting to use the methods reported here to study the difference between EC and PC in their interaction with graphite anode and the SEI formation. It is probably much more relevant to the battery field than DX on graphite. The authors are recommended to provide data and discussion on PC-based electrolytes.

REV1_R3: We thank Reviewer for another good point. We systematically measured the solvent co-intercalation dependent nanoscale SEI formation in different electrolytes with various solvents, including EC and PC and their mixture with co-solvent such as DEC, DMC, etc. One of the examples of solvent (PC) co-intercalation induced damage are shown in Figure R1 below, in which the carbon atomic steps exfoliation (Figure R1a) in the basal plane,

as well as the damages to the HOPG model samples at the sectional plane (Figure R1b)⁹, are the two main obstacles for performing further detailed high-resolution *in-situ/operando* electrochemical AFM measurements^{10, 11}.

Figure R1. (a) Carbon layers peeling off from sample surface caused by solvent co-intercalation. (b) *Operando* electrochemical optical microscopy captures the damage of the sectional area in the graphite model sample during the lithiation in PC-based electrolyte.

To further address this question, we added **Section S5 and Figure S5.1** to the revised SI (see detail discussion below) and references to these data in the main manuscript in **P9, lines 3-7**.

Figure S5.1 summarized the different solvent co-intercalation effects on our model sample. The electrolyte containing PC solvent cause serious carbon layer expansion/exfoliation due to PC co-intercalation (Figure S5.1a); electrolyte containing DEC co-solvent can also co-intercalate and slightly weak the Van Der Waals interaction of carbon layers, generating many nano-blisters/bubbles trapped inside graphite^{12, 13} (Figure S5.1b), which is detrimental for the mechanical property quantification of later formed SEI layers; In EC/DMC mixed electrolyte, the solvent co-intercalation and sequential decomposition effectively sealed the carbon step edges¹⁴ and form carbon edge wrappings at initial lithiation stage (Figure S5.1c), preventing the further graphite exfoliations/delamination. EC/DMC passivated HOPG surface can thereby serve as an idea “solid-substrate” for the study of nanoscale mechanical properties of SEI layers. The detailed solvent/co-solvent co-intercalation behaviour will be discussed in our next work. Besides, since this study focuses on the effects of solvent co-intercalation and solvation structure on SEI formation, we therefore

selected the weakly solvated solvent which has distinct solvation structures with lithium ions^{15, 16, 17}, to eliminate the solvent co-intercalation resulted in carbon layer exfoliation and sample damages¹⁸.

Fig. S5.1 Three-dimensional surface topography of HOPG surface with carbon atomic steps before and after the solvent co-intercalation/decomposition in 1M LiPF₆ (a) PC, (b) EC: DEC=1:1 (v/v) and (c) EC: DMC=1:1 (v/v) electrolytes.

Reviewer #2 (Remarks to the Author):

SEI is a critical but complex topic in battery research, and it is still challenging to fully understand through both computational and experimental investigation. Developing any new methodology, therefore, is very important to the field. This work presents a new method using both electrochemical atomic force microscopy and 3D nano-rheology microscopy to identify the chemical composition of SEI and illustrate its early formation process. However, I feel that some experiments are not explained very clearly, and the MD simulation part doesn't provide much useful support or insight. Here are some detailed questions for the authors to consider:

Rev2_Response: We thank the reviewer's positive comments and constructive suggestions. We have expanded the discussion of several key results in the main manuscript and added in the revised supplementary information (SI) the expanded discussions of XPS measurements, as well as MD simulation details supporting the interfacial structure inside the EDL:

XPS

SI, Section 7, P15, lines 6-24.

SI, Section 7, Figure S7.1 with captions.

MD simulation

SI, Page 2, lines 15-39.

SI, Section 10, Figure S10.2, Figure S10.3

SI, Section 10, P21, lines 8-33

please see the point-by-point response to the questions 1-10 below.

(1) On Page 3, please add which electrolyte systems to study in this work before the RESULTS section.

Rev2_R1: We thank the Reviewer's question. The electrolyte systems are now specified before the RESULTS section and the cross-comparison of different electrolyte systems was further added in the revised manuscript.

Page 3, lines 32-34, "We use a matrix of two morphologically dissimilar but chemically identical surfaces of typical carbon electrode material (basal and edge graphene planes) and two solvent-electrolyte systems, polar solvent (Ethylene Carbonate, EC, mixed with Dimethyl Carbonate, DMC) and non-polar (1,4 Dioxane, DX) solvent, to get direct insight into the atomistic pictures for the underlying influence of solid-liquid interfacial nanostructure on the initial SEI formation."

(2) Fig 1K, what is the peak near 160 nm?

Rev2_R2: We thank Reviewer's question. The peak near 160 is one of the wrapping carbon steps after the solvent co-intercalation/decomposition. We replotted the topography images in three-dimensional as shown in **SI Section 4 Fig. S4.5** (see below), the line structures observed on the sample surface during the charge/discharge cycles are the SEI formed and accumulated at the carbon step edge^{14, 19}. In other words, the peak near 160 nm (Figure 1j in the main text) are the electrolyte decomposition products preferentially accumulated at one of the step edges. To better explain this SEI decomposition-induced carbon step edge expansion/wrapping, we specify this section profiles in the legend and add the below 3D presented topography in the revised supplementary information.

Page 4, lines 8-10, starting with "(i) and (j) Cross-section profiles ..." and amended Figure 1i and 1j.

SI Section 4, Figure S4.5 with captions.

Fig. S4.5 (a) Three-dimensional topography images of graphite electrode surface with carbon atomic steps captured at different lithiation states. (b) High-magnification images of carbon steps during the lithiations. Three pictures correspond to the green, blue and purple dash lined square areas at different lithiation states in Fig. S4.5a. (c) The schematic model of SEI accumulation at a carbon atomic step.

(3) Page 6, Please add a clear discussion to explain the distinct SEI structures between the edge and basal planes via XPS studies, since this was mentioned at the beginning of the next part.

Rev2_R3: We thank the Reviewer for a valid suggestion. The detailed XPS discussion was added in the main manuscript:

Page 8, lines 13-31, starting with “The SEI chemical structures were investigated by X-ray photoelectron spectroscopy...” and in SI

(4) Overall, I feel the figures are all too small. Some letters and features are hard to read.

Rev2_R4: We agree with this suggestion. We redesigned the layout of small figures and increased the image resolutions and fonts in the revised manuscript.

(5) Figure 4c is kind of misleading. The authors always mention the EC-Li coordination for EC-based electrolyte, which is also demonstrated in Figure 4c that only the EC coordination appears in the inner-most layers next to the anode surface, it should be noted that the real electrolyte used here is 1 M LiPF₆ in EC/DMC (please also explain the molar or volume ratio between EC/DMC on page 4). It is expected that DMC also coordinates to Li, probably even at a larger number depending on the EC: DMC ratio, because the literature (Fundamental Research 1 (2021) 393-398) suggests a stronger Li-DMC binding energy than Li-EC. This should be able to validate through MD simulations.

Rev2_R5: We thank Reviewer for pointing out the inaccurate solvation structure schematic and bringing up this useful reference and insights to our attention. We specified the ratio EC: DMC=1:1 (v:v) and corrected the solvation structure schematic in Figure 4 according to our additional Raman characterization and MD simulation results in **SI Section 11**. The results are consistent with the reported Raman/IR/NMR/SIMS experiments^{20, 21, 22, 23, 24} and MD/DFT simulation results^{25, 26, 27} which suggest that $(EC)_{3-4}(DMC)_{1-2}Li^+$ is the most stable solvates, namely, a lithium-ion is most probably solvated by 3~4 EC molecules and 1~2 DMC molecule inside the first solvation sheath in 1 M LiPF₆ EC: DMC=1:1 (v:v). Making it consistent with the literature references including (Fundamental Research 1 (2021) 393-398) as Ref 74 of the revised manuscript. To address this, we made appropriate correction and added the following detailed Raman characterizations and MD simulation results in the revised manuscript and SI.

Page 11, lines 6-9, starting with “The different solvation structures ...”

Page 12, Figure 5b and 5b’ with caption “Raman spectroscopy characterization of two electrolyte bulk solvation structures ...”

SI Page 21, lines 8-33, starting with “Apart from the density distribution profiles along the...”

SI Section 11, Page 23, lines 1-37, starting with “We performed Raman spectroscopy measurements...”

Figures S11.1, S11.2 with captions (see below).

Fig. S11.1 (a) and (b) Raman spectra of pure EC, DMC and 1M LiPF₆ in EC: DMC=1:1 electrolyte. (c) Radial distribution function of Li-O (EC), Li-O (DMC) and Li-F from MD simulation results in 1M LiPF₆ in EC: DMC=1:1 electrolyte. ($g(r)$ functions inside and outside (bulk) the EDL were calculated by $0 < Z_{EDL} < 2$ nm, and $Z_{Bulk} > 2$ nm according to the experiment measured force-distance curves)

Fig. S11.2 (a) Raman spectra of pure DX, 0.5 and 1M LiFSI in DX electrolyte. (b) Deconvoluted S-N-S stretch band (c) (d) RDF of Li-O (DX) and Li-O (FSI) from MD simulation results in 1M LiFSI in DX electrolyte. ($g(r)$ functions inside and outside (bulk) the EDL were calculated by $0 < Z_{EDL} < 2$ nm, and $Z_{Bulk} > 2$ nm according to the experiment measured force-distance curves)

(6) The XPS in Figure S11 e and h also shows the Li-F SEI component on both basal and sectional planes, is this due to the decomposition of anion? The relative Li-F peak is higher on the basal plane, why?

Rev2_R6: We gratefully acknowledge the comment. We believe the Li-F component is mainly from the hydrolysis and electrochemical decomposition of PF_6^- anion^{14, 28}. In our graphite model sample, the main SEI components formed in the basal plane derive from the heterogeneous transfer of electrons directly to the salt anions and solvents. This heterogeneous transfer of electrons results in electrolyte decomposition precipitates²⁹, including a large amount of anion decomposition products which contain more LiF. By contrast, the SEI layer formed on the edge plane mainly derives from the decomposition of co-intercalated EC solvents which results in the accumulation of lithium ethylene di-carbonate (LEDC), or lithium ethylene mono-carbonate (LEMC)³⁰ in the very end of intercalation active edge planes²⁹. The accumulation of these organic species, instead of anion decomposition precipitations, forms the main SEI components on the sample sectional plane. Therefore, the sectional plane shows lower Li-F peak intensity compared to the basal plane.

(7) Why the similar Figures in Figure S11 are not measured for the DX-based electrolyte?

Rev2_R7: We thank the Reviewer for noticing this. The SEI formed on two planes in the DX-based electrolyte are both anion-derived SEI (independent of graphite crystal orientations) that has similar chemical components, and thereby were not discussed in detail. The detailed discussion and Li1s, S2p, N1s, O1s, C1s, and F1s high-resolution spectra of the basal plane and sectional plane SEI formed in DX electrolyte has been added to **SI Figure S7.1** in the revised manuscript.

With discussion in

SI Section 7, Page 15, lines 6-24 starting with “Fig. S7.1a shows the full survey...”

(8) Are insets figures in figures 4e and 4f from MD simulations? Li should also be highlighted.

Rev2_R8 We acknowledge this comment from Reviewer. We highlighted the Lithium-ions in the insets in Figure 4e and 4f and add these two MD simulations results in the SI **SI, Section 11, Figure S10.2.**

(9) It is mentioned on page 11 that “The DX solvent molecules rarely exist inside the inner Helmholtz layer” which is not supported by the inset in Fig 4f.

Rev2_R9: We thank reviewer’s useful comment, the wrong statement has been changed to “due to the weak coordination effect between DX and lithium ... FSI⁻ still dominates the first solvation sheath inside the EDL” which is supported by the Raman and MD simulation in **Figure S11.1.**

The amendments are made in the revised manuscript

Page 12, lines 7-11.

(10) MD simulation methods are not described clearly which is not enough for others to reproduce the work. The MD results are also not reported and explained at all, not sure what is the purpose of the MD part. The MD results should provide detailed and clear information on the interfacial structure of the electrolyte. If the results were not discussed, this part should be removed.

Rev2_R10: We thank Reviewer for another useful comment. To provide detailed and clear information on the interfacial structure of the electrolyte, the MD simulation details are added in the revised SI METHOD section, as well as the information on the interfacial structure of the electrolyte is added in the revised SI Section 10.

MD detailed description.

SI, Page2, lines 15-39.

Electrolyte solvation structure

SI, Page 21, lines 8-33 Starting with “Apart from the density distribution...” and **Fig. S10.3** below.

Fig. S10.3 Radial distribution function (RDF) and coordination numbers of Lithium in (a, b) EC and (c, d) DX based electrolyte inside and outside of EDL under the negatively charged electrode surface. ($g(r)$ functions inside and outside (bulk) the EDL were calculated by $0 < Z_{EDL} < 2$ nm, and $Z_{Bulk} > 2$ nm according to the experiment measured force-distance curves) (e, f) Three types of Li-FSI solvation structures inside EDL of DX based electrolyte. Small brown ball is Li⁺, large-side spherical structure ions are FSI⁻, stick structure are solvent molecules/ions in the background.

The authors would like to thank the Reviewers and Editorial board again for the detailed and thoughtful comments on the manuscript and hope that we were able to address the questions and suggestions in full.

We will provide both the unmarked manuscript and SI as well as marked version with line numbers and changes and additions highlighted.

With kind regards,

Oleg Kolosov (on behalf of all the authors).

References

1. Song J, Zhou Y, Padture NP, Huey BD. Anomalous 3D nanoscale photoconduction in hybrid perovskite semiconductors revealed by tomographic atomic force microscopy. *Nature Communications* **11**, 3308 (2020).
2. v Cresce A, Russell SM, Baker DR, Gaskell KJ, Xu K. In situ and quantitative characterization of solid electrolyte interphases. *Nano letters* **14**, 1405-1412 (2014).
3. Zhang Z, *et al.* Characterizing Batteries by In Situ Electrochemical Atomic Force Microscopy: A Critical Review. *Advanced Energy Materials* **11**, (2021).
4. Zhang Z, Smith K, Jarvis R, Shearing PR, Miller TS, Brett DJL. Operando Electrochemical Atomic Force Microscopy of Solid–Electrolyte Interphase Formation on Graphite Anodes: The Evolution of SEI Morphology and Mechanical Properties. *ACS Applied Materials & Interfaces* **12**, 35132-35141 (2020).
5. Zhang Z, *et al.* Capturing the swelling of solid-electrolyte interphase in lithium metal batteries. *Science* **375**, 66-70 (2022).
6. Gao Y, *et al.* Unraveling the mechanical origin of stable solid electrolyte interphase. *Joule* **5**, 1860-1872 (2021).
7. Zheng J, *et al.* 3D visualization of inhomogeneous multi-layered structure and Young's modulus of the solid electrolyte interphase (SEI) on silicon anodes for lithium ion batteries. *Physical Chemistry Chemical Physics* **16**, 13229-13238 (2014).
8. Gu Y, *et al.* Designable ultra-smooth ultra-thin solid-electrolyte interphases of three alkali metal anodes. *Nature communications* **9**, 1339 (2018).
9. Xu K. Whether EC and PC Differ in Interphasial Chemistry on Graphitic Anode and How. *Journal of The Electrochemical Society* **156**, (2009).
10. Jeong S-K, Inaba M, Iriyama Y, Abe T, Ogumi Z. AFM study of surface film formation on a composite graphite electrode in lithium-ion batteries. *Journal of Power Sources* **119-121**, 555-560 (2003).
11. Song H-Y, Jeong S-K. Investigating continuous co-intercalation of solvated lithium ions and graphite exfoliation in propylene carbonate-based electrolyte solutions. *Journal of Power Sources* **373**, 110-118 (2018).
12. Jeong S-K, Inaba M, Abe T, Ogumi Z. Surface Film Formation on Graphite Negative Electrode in Lithium-Ion Batteries: AFM Study in an Ethylene Carbonate-Based Solution. *Journal of The Electrochemical Society* **148**, (2001).

13. Jeong S-K, Inaba M, Mogi R, Iriyama Y, Abe T, Ogumi Z. Surface Film Formation on a Graphite Negative Electrode in Lithium-Ion Batteries: Atomic Force Microscopy Study on the Effects of Film-Forming Additives in Propylene Carbonate Solutions. *Langmuir* **17**, 8281-8286 (2001).
14. Liu T, *et al.* In situ quantification of interphasial chemistry in Li-ion battery. *Nature nanotechnology* **14**, 50-56 (2019).
15. Ding JF, *et al.* Non - Solvating and Low - Dielectricity Cosolvent for Anion - Derived Solid Electrolyte Interphases in Lithium Metal Batteries. *Angewandte Chemie International Edition*, (2021).
16. Jiang LL, Yan C, Yao YX, Cai W, Huang JQ, Zhang Q. Inhibiting Solvent Co - Intercalation in a Graphite Anode by a Localized High - Concentration Electrolyte in Fast - Charging Batteries. *Angewandte Chemie International Edition* **60**, 3402-3406 (2020).
17. Yao YX, *et al.* Regulating Interfacial Chemistry in Lithium - Ion Batteries by a Weakly Solvating Electrolyte**. *Angewandte Chemie International Edition* **60**, 4090-4097 (2020).
18. Ming J, *et al.* New Insights on Graphite Anode Stability in Rechargeable Batteries: Li Ion Coordination Structures Prevail over Solid Electrolyte Interphases. *ACS Energy Letters* **3**, 335-340 (2018).
19. Yang K, *et al.* Revealing the anion intercalation behavior and surface evolution of graphite in dual-ion batteries via in situ AFM. *Nano Research* **13**, 412-418 (2020).
20. Morita M, Asai Y, Yoshimoto N, Ishikawa M. A Raman spectroscopic study of organic electrolyte solutions based on binary solvent systems of ethylene carbonate with low viscosity solvents which dissolve different lithium salts. *Journal of the Chemical Society, Faraday Transactions* **94**, 3451-3456 (1998).
21. Yamada Y, Sagane F, Iriyama Y, Abe T, Ogumi Z. Kinetics of Lithium-Ion Transfer at the Interface between Li_{0.35}La_{0.55}TiO₃ and Binary Electrolytes. *The Journal of Physical Chemistry C* **113**, 14528-14532 (2009).
22. Bogle X, Vazquez R, Greenbaum S, Cresce A, Xu K. Understanding Li(+)-Solvent Interaction in Nonaqueous Carbonate Electrolytes with (17)O NMR. *J Phys Chem Lett* **4**, 1664-1668 (2013).
23. Seo DM, Reininger S, Kutcher M, Redmond K, Euler WB, Lucht BL. Role of Mixed Solvation and Ion Pairing in the Solution Structure of Lithium Ion Battery Electrolytes. *The Journal of Physical Chemistry C* **119**, 14038-14046 (2015).
24. Zhang Y, *et al.* Investigation of Ion-Solvent Interactions in Nonaqueous Electrolytes Using in Situ Liquid SIMS. *Analytical chemistry* **90**, 3341-3348 (2018).
25. Borodin O, Olguin M, Ganesh P, Kent PR, Allen JL, Henderson WA. Competitive lithium solvation of linear and cyclic carbonates from quantum chemistry. *Phys Chem Chem Phys* **18**, 164-175 (2016).
26. Skarmoutsos I, Ponnuchamy V, Vetere V, Mossa S. Li⁺ Solvation in Pure, Binary, and Ternary Mixtures of Organic Carbonate Electrolytes. *The Journal of Physical Chemistry C* **119**, 4502-4515 (2015).
27. Ponnuchamy V, Mossa S, Skarmoutsos I. Solvent and Salt Effect on Lithium Ion Solvation and Contact Ion Pair Formation in Organic Carbonates: A Quantum Chemical Perspective. *The Journal of Physical Chemistry C* **122**, 25930-25939 (2018).

28. Xu K. Electrolytes and interphases in Li-ion batteries and beyond. *Chem Rev* **114**, 11503-11618 (2014).
29. An SJ, Li J, Daniel C, Mohanty D, Nagpure S, Wood DL. The state of understanding of the lithium-ion-battery graphite solid electrolyte interphase (SEI) and its relationship to formation cycling. *Carbon* **105**, 52-76 (2016).
30. Wang L, *et al.* Identifying the components of the solid–electrolyte interphase in Li-ion batteries. *Nature chemistry* **11**, 789-796 (2019).

Peer review comments, second round –

Reviewer #1 (Remarks to the Author):

The revision has satisfied my concerns in the previous review. I recommend the manuscript to be published without further revision.

Reviewer #2 (Remarks to the Author):

I have a few more questions regarding the revised manuscript.

Regarding MD simulations:

- (1) How do you decide the simulation box size and number of electrolyte species?
- (2) There is no description to explain how the -1 V vs. PZC was simulated (Figure S10.3).
- (3) Are 3D periodic boundary conditions applied? According to Figure S10, it appears that a vacuum space is added to separate the two electrodes. If the simulation uses 3D periodic boundary conditions, the 5 nm vacuum space is too short for separating two oppositely charged electrodes. It is recommended to use the vacuum length of twice the electrolyte box to validate your results again.

Page 12, lines 319-326, here, the authors discuss the different existence of the PF6 and FSI anion in Li-solvation structures near the electrode due to different lithium solvent coordination abilities. However, the two electrolyte systems studied here have both different solvents (EC/DMC vs. DX) and lithium salts (LiPF6 vs. LiFSI). It is uncertain whether different anion chemistries can also lead to different interfacial chemistries in the same organic solvent electrolyte.

REVIEWER'S COMMENTS

We thank the Reviewers for the positive evaluation of the manuscript and helpful comments. Please find attached the detailed response to the comments raised by the Reviewer 2.

Reviewer #1 (Remarks to the Author):

The revision has satisfied my concerns in the previous review. I recommend the manuscript to be published without further revision.

Response: We thank the Reviewer for the positive feedback on the revised manuscript.

Reviewer #2 (Remarks to the Author):

I have a few more questions regarding the revised manuscript.

Regarding MD simulations:

(1) How do you decide the simulation box size and number of electrolyte species?

Response: We thank the Reviewer for the in-depth and constructive comments on MD simulation and are answering the questions raised in details below.

The size of the simulation box and the number of electrolyte species were determined by ensuring the electrostatic crosstalk effect between two carbon electrodes can be eliminated, meanwhile, enough molecules can also be used for the statistical analysis, such as coordination radius distribution functions (RDF) and interfacial ion distribution functions.

To be more specific, first of all, a 10 nm distance between two electrodes was chosen such that the forces on the molecules in the middle of the simulation box were statistically the same as those in the bulk simulations without electrodes^{1,2}. Additionally, in every simulation result, we also examined the solvation structures of lithium-ions in the polarized bulk electrolyte by RDF (Figure S10.3 b and S10.3 d of supplementary information, SI), which is similar to that of bulk electrolyte under the un-polarized state (Figure S11.1 c and S11.1 d of SI). Namely, the half distance of the two electrodes was controlled to be larger than the Debye length of EDL (around 2-3 nm according to the force curve experiments). Besides, the electrode area size of around 5.1×5.1 nm² was chosen to ensure the continuity of the in-plane periodic structure of carbon hexagonal rings in two electrodes.

Once the above simulation box size is determined, the number of each electrolyte species should be enough for the statistical analysis of EDL structures and lithium-ions' solvation structures. The detailed numbers of each species can be calculated according to the densities, concentration and volume ratio of solvents in the electrolyte as discussed below. We add the detailed statement in this in the revised SI (lines 34-41, Page 2 and lines 3-27, Page 3).

“The detailed numbers of each species can be calculated according to the densities, concentration and volume ratio of solvents in the electrolyte. The volume of simulation box is around 2.60×10^{-19} cm³. For the 1M concentration electrolytes, the simulation box contains about 2.60×10^{-22} mole salt molecules, which corresponds to about 156 LiPF₆ or LiFSI. Since the electrolyte is dilute, we ignored the

volume changes of the simulation box before and after adding salt. (This results on the error of less than 1% for these values, see the results in Table R1 below) The volume ratio of EC: DMC is 1:1, therefore taking the density of $1.33\pm 0.01 \text{ g/cm}^3$ for EC and $1.07\pm 0.01 \text{ g/cm}^3$ for DMC at room temperature, the mass in the simulation box will be around for EC $1.72\text{-}1.74 \times 10^{-19} \text{ g}$ and for DMC $1.38\text{-}1.40 \times 10^{-19} \text{ g}$, corresponding to the numbers of around 1175-1189 and 920-936 for EC and DMC molecules, respectively. The molecule numbers within this range are in an acceptable error range. Similarly, the number of DX in the simulation box and is around 1638, by using a room temperature density value of 1.03 g/cm^3 .

For a more precise estimation using the final density of the commercial electrolyte (1.3634 g/mL for LiPF₆ in EC: DMC=1:1 (v:v) electrolyte), the number of LiPF₆ salt, EC, and DMC are determined as around 156.52, 1195.63 and 935.63, respectively (see the detailed calculation parameter in Table R1). The numbers we used for LiPF₆, EC, and DMC are 156, 1187, and 934 respectively, which results in the error smaller than 1% using this more precise estimation.”

Table R1 Detailed parameters for determining the molecule numbers in the simulation box by the overall density of LiPF₆ in EC: DMC=1:1 (v:v) electrolyte

Species	Mass in the simulation box (g)	Molecules weight (g/mol)	Numbers in the simulation box
LiPF ₆	3.95×10^{-20}	151.905	156.52
EC	1.749×10^{-19}	88.062	1195.63
DMC	1.400×10^{-19}	90.078	935.63
Total mass (g)	3.544×10^{-19}		

Note:

Salt concentration: 1 M mol/mL

Volume ratio of EC: DMC=1:1, therefore the mass ratio $\frac{m_{EC}}{m_{DMC}} = \frac{\rho_{DMC}}{\rho_{EC}} = \frac{1.07}{1.33} \approx 0.80$

Electrolyte density: 1.3634 g/mL (see: <https://www.sigmaaldrich.com/GB/en/product/aldrich/809357>)

Volume of box: $2.60 \times 10^{-19} \text{ cm}^3$

Total mass: $3.544 \times 10^{-19} \text{ g}$

Avogadro constant: 6.02×10^{23}

(2) There is no description to explain how the -1 V vs. PZC was simulated (Figure S10.3).

Response: We thank the Reviewer for this comment. To address this, we add the following simulation details below in the revised methods (lines 21-29, Page 2 in the revised SI).

“In EC-based electrolyte, the negative charge ($\Delta Q=0.0036 \text{ e}$) is equally added to each carbon atom in one of the electrodes to generate the polarization field, meanwhile, the other electrode was positively charged by the same amount of charges. According to the parallel plate capacitance model, the voltage can be estimated by the following equation,

$$V = \frac{\Delta Q}{C} = \frac{\Delta Q d}{k \epsilon_0 A}$$

Where d is the distance between two electrodes, k is the dielectric constant of electrolyte, A is the averaged area of each carbon atom occupying in a unit cell and ϵ_0 is the permittivity of vacuum. The dielectric constant of EC- and DX-based electrolytes are around $\sim 25^3$ and $\sim 2.2^4$, respectively. Therefore, to simulate a voltage bias value of $V \approx -1$ V on the carbon electrodes, we added $\Delta Q=0.0036$ e negative charges to each carbon on the electrode for EC-based electrolyte system, and $\Delta Q=0.00032$ e negative charges to each carbon on the electrode for DX-based electrolyte system.”

(3) Are 3D periodic boundary conditions applied? According to Figure S10, it appears that a vacuum space is added to separate the two electrodes. If the simulation uses 3D periodic boundary conditions, the 5 nm vacuum space is too short for separating two oppositely charged electrodes. It is recommended to use the vacuum length of twice the electrolyte box to validate your results again.

Responses: We thank Reviewer for this useful comment that allows us to further clarify the MD simulation parameter selection.

First, we confirm that 3D periodic boundary conditions are applied with appropriate line added in the SI (**line 42, 43 page 2**), As required by Reviewer, all simulation results were verified again using a larger vacuum length (10 nm) as shown in Figures R1a-b. The RDF results confirm that the solvation structure and coordination numbers inside the EDL and bulk electrolyte for EC-based electrolyte (Figures R1c-d) and DX-based electrolyte (Figures R1e-f) are both similar to the results calculated using the 5 nm vacuum distance. It is also worth noting that the coordination number of EC in the bulk electrolyte slightly decreased to about 3 after the vacuum length is increased, which is more consistent with previous reports^{5, 6}. Importantly, according to the RDF results, the EC/DMC solvents and FSI⁻ anion are still dominating in the first solvation shell for EC-based and DX-based electrolytes, respectively. Besides, Figures R1g and R1h show that the distributions of cation, anion and solvent along the out-of-plane direction follow the same trend as the results in Figure 4 in the original main manuscript.

To address this comment, we add the details of new simulation box in **Figure S10.1 of SI** and amended simulation results in the revised manuscript.

Figure R1

Simulation results using a larger vacuum space (10 nm). Visual representation of simulation box for (a) 156 LiPF₆, 1187 EC, and 934 DMC and 934 and (b) 156 LiFSI/1638 DX confined by two carbon (001). The RDF of Li-O (EC), Li-O (DMC), and Li-F inside the (c) EDL and (d) in the bulk electrolyte of EC-based electrolyte. The RDF of Li-O (DX), Li-O (FSI), and Li-N inside the (e) EDL and (f) in the bulk electrolyte of DX-based electrolyte. The distributions of cation, anion and solvent along the out-of-plane direction in (g) EC-based electrolyte and (h) DX-based electrolytes.

Page 12, lines 319-326, here, the authors discuss the different existence of the PF₆ and FSI anion in Li-solvation structures near the electrode due to different lithium solvent coordination abilities. However, the two electrolyte systems studied here have both different solvents (EC/DMC vs. DX) and lithium salts (LiPF₆ vs. LiFSI). It is uncertain whether different anion chemistries can also lead to different interfacial chemistries in the same organic solvent electrolyte.

Responses: We agree with the Reviewer that FSI and PF₆ may result in different interfacial chemistries. Nevertheless, according to the results of this study, and supported by the additional experiments described below and included in the SI, we believe that the competitive coordination effect of solvent and anion toward lithium-ion, which determines the lithium-ion solvation structures, is more important than the sole effect of different anion (PF₆ and FSI) chemistries in terms of solvent or anion preferential decomposition^{7, 8}. The SEI formation in EC/DMC with different anions (PF₆/FSI) were previously discussed in the

supplementary information (Figure S8.1) of the original manuscript. The effective modulus and viscosity measurements in Figure S8.1 indicates that the SEI layer formed in 1M LiPF₆ in EC/DMC and 1M LiFSI in EC/DMC on lithium metal anode have similar mechanical properties, which are both derived from the solvent decomposition-induced soft, predominantly organic SEI components.

Below are detailed explanations based on the controlled experiments that are addressing this comment, that are now added in the revised SI, section 8 (**line 31, page 19 through line 18, page 21**) and a comment in the main manuscript (**lines 34-36, Page 13**).

1. LiPF₆/LiFSI salt in the same commercial EC/DMC solvent system.

“The above considerations suggest (and are supported by the explanations below based on the controlled experiments) that the competitive coordination effect of solvent and anion toward cations, which determines the lithium-ion solvation structures, is more important than the sole effect of different anion (PF₆ and FSI) chemistries in terms of electrolyte decomposition paths (solvent or anion dominated).

This can be further explained by the Raman spectra of LiFSI in EC/DMC as shown in Figure R2a. The strong EC-Li coordination peaks, at around 723 cm⁻¹ and 915 cm⁻¹, are observed in LiFSI in EC/DMC electrolytes at concentrations ranging from 0.1 to 6 M, indicating the strong coordination ability of high dielectric constant EC solvent compared with FSI anions⁹. This solvent preferential coordination is similar to the 1M LiPF₆ in EC/DMC as proved in the Raman spectrum in Figure 5b of the main manuscript. Additionally, comparing Figure R2b with R2c, the S-F-S stretch peak derived from the aggregated Li-FSI coordination at around 747 cm⁻¹ does not exist in EC/DMC based electrolyte, confirming that the lithium ions in 1M LiFSI mixed with EC/DMC electrolyte mainly form uncoordinated free ion and separated ion-pairs, rather than the ion aggregates as in weakly solvating DX-based electrolyte. In conclusion, Li-ions in 1M LiFSI in EC/DMC are mainly solvated by solvent molecules in the first solvation shell, similar to 1M LiPF₆ in EC/DMC electrolyte, but different from 1M LiFSI in DX.

As a result, the SEI measurement by AFM (Figure R1c) found that, similar to the SEI structures formed in 1M LiPF₆ in EC/DMC in Figure 3i of main manuscript, a soft organic-rich SEI layer in the sample section and stiff inorganic-rich SEI layer on the basal plane are also formed in the sample cycled in 1M LiFSI in EC/DMC. We, therefore, suggested that it is the solvation structure (competition effects of solvents and anions), rather than the anion species, that dominates the preferential electrolyte decomposition process in these electrolyte systems^{10, 11}.

Figure R2 (a) Raman spectra of LiFSI in EC: DMC=1:1 electrolyte with different salt concentrations. Raman spectra showing the FSI anions coordination statuses in (b) 1M LiFSI in EC: DMC=1:1 and (c) 1M LiFSI in DX electrolyte. Surface morphology, peak-force error, and DMT nano-mechanical modulus of SEI layers formed in (d) 1M and (e) 6M LiFSI in EC: DMC=1:1 electrolyte.

More significantly, another proof of the importance of solvation structure is as follows: as shown in Figure R2e, in a relatively high concentration (6M) LiFSI in EC/DMC electrolyte, the SEI layer becomes a uniform stiff layer on both sample section and basal plane. This is because when the concentration reaches 6M (LiFSI: EC: DMC \approx 1:1.3:0.9), almost every oxygen atom in EC is coordinating with Li-ions according to the Raman spectrum in Figure R2a, but these EC is not enough to “surround” all Li-ions and screen the electrostatic charges. In this case, FSI anions inevitably participate in the first solvation shell. This increased the possibility of anion reduction on the electrode surface and results in more anion-derived SEI interfacial chemistries, similar to the weakly solvating electrolyte using DX solvent.

2. LiPF₆/LiFSI salt in the non-polar DX solvent system:

To prepare the electrolyte with weak cation-solvent interaction, we intentionally chose LiFSI to mix with DX, rather than LiPF₆. This is because that LiPF₆ is barely dissolved by most non-polar solvents, such as benzene and DX, but FSI can be dissolved due to its entropy effect induced by large size¹². These non-polar, low-dielectric constant, and low-dissociation capability solvents were carefully selected in other recent studies^{13, 14} to explore the effect of solvation structure (anion/solvent coordination abilities) and have been demonstrated their effectiveness on SEI interfacial chemistry modifications.

Overall, we believed that the competition between the anion and solvents in the first solvation shell can be tailored by changing the strongly/weakly solvating solvent species and salt concentration, which plays a more important role in determining the SEI formations compared with anion chemistries. By modulating the dominating coordinators with lithium-ions, one can control the SEI interfacial chemistries by guiding the electrolyte decomposition paths.”

The authors would like to thank the Reviewers and the Editorial board for the detailed and thoughtful comments on the manuscript and hope that we were able to address the questions and suggestions in full.

We will provide both the unmarked manuscript and SI as well as marked version with line numbers and changes and additions highlighted.

With kind regards,

Oleg Kolosov (on behalf of all the authors).

References

1. Xing, L.; Vatamanu, J.; Borodin, O.; Smith, G. D.; Bedrov, D., Electrode/Electrolyte Interface in Sulfolane-Based Electrolytes for Li Ion Batteries: A Molecular Dynamics Simulation Study. *The Journal of Physical Chemistry C* **2012**, *116* (45), 23871-23881.
2. Vatamanu, J.; Borodin, O.; Smith, G. D., Molecular Dynamics Simulation Studies of the Structure of a Mixed Carbonate/LiPF₆ Electrolyte near Graphite Surface as a Function of Electrode Potential. *The Journal of Physical Chemistry C* **2011**, *116* (1), 1114-1121.
3. Yao, N.; Chen, X.; Shen, X.; Zhang, R.; Fu, Z.-H.; Ma, X.-X.; Zhang, X.-Q.; Li, B.-Q.; Zhang, Q., An Atomic Insight into the Chemical Origin and Variation of the Dielectric Constant in Liquid Electrolytes. *Angewandte Chemie International Edition* **2021**, *60* (39), 21473-21478.
4. Yao, Y. X.; Chen, X.; Yan, C.; Zhang, X. Q.; Cai, W. L.; Huang, J. Q.; Zhang, Q., Regulating Interfacial Chemistry in Lithium - Ion Batteries by a Weakly Solvating Electrolyte**. *Angewandte Chemie International Edition* **2020**, *60* (8), 4090-4097.
5. Skarmoutsos, I.; Ponnuchamy, V.; Vetere, V.; Mossa, S., Li⁺ Solvation in Pure, Binary, and Ternary Mixtures of Organic Carbonate Electrolytes. *The Journal of Physical Chemistry C* **2015**, *119* (9), 4502-4515.
6. Borodin, O.; Olguin, M.; Ganesh, P.; Kent, P. R.; Allen, J. L.; Henderson, W. A., Competitive lithium solvation of linear and cyclic carbonates from quantum chemistry. *Phys Chem Chem Phys* **2016**, *18* (1), 164-75.
7. Cheng, H.; Sun, Q.; Li, L.; Zou, Y.; Wang, Y.; Cai, T.; Zhao, F.; Liu, G.; Ma, Z.; Wahyudi, W.; Li, Q.; Ming, J., Emerging Era of Electrolyte Solvation Structure and Interfacial Model in Batteries. *ACS Energy Letters* **2022**, *7* (1), 490-513.
8. Yu, D.; Zhu, Q.; Cheng, L.; Dong, S.; Zhang, X.; Wang, H.; Yang, N., Anion Solvation Regulation Enables Long Cycle Stability of Graphite Cathodes. *ACS Energy Letters* **2021**, *6* (3), 949-958.
9. Uchida, S.; Ishikawa, M., Lithium bis(fluorosulfonyl)imide based low ethylene carbonate content electrolyte with unusual solvation state. *Journal of Power Sources* **2017**, *359*, 480-486.
10. Lei, S.; Zeng, Z.; Liu, M.; Zhang, H.; Cheng, S.; Xie, J., Balanced solvation/de-solvation of electrolyte facilitates Li-ion intercalation for fast charging and low-temperature Li-ion batteries. *Nano Energy* **2022**, *98*, 107265.

11. Zhang, L.; Chen, Y., Electrolyte solvation structure as a stabilization mechanism for electrodes. *Energy Materials* **2021**, *1* (1), 100004.
12. Chen, X.; Zhang, Q., Atomic Insights into the Fundamental Interactions in Lithium Battery Electrolytes. *Accounts Chem. Res.* **2020**, *53* (9), 1992-2002.
13. Fan, X.; Ji, X.; Chen, L.; Chen, J.; Deng, T.; Han, F.; Yue, J.; Piao, N.; Wang, R.; Zhou, X.; Xiao, X.; Chen, L.; Wang, C., All-temperature batteries enabled by fluorinated electrolytes with non-polar solvents. *Nature Energy* **2019**, *4* (10), 882-890.
14. Ding, J. F.; Xu, R.; Yao, N.; Chen, X.; Xiao, Y.; Yao, Y. X.; Yan, C.; Xie, J.; Huang, J. Q., Non - Solvating and Low - Dielectricity Cosolvent for Anion - Derived Solid Electrolyte Interphases in Lithium Metal Batteries. *Angewandte Chemie International Edition* **2021**.

Peer review comments, third round –

Reviewer #2 (Remarks to the Author):

I only have two comments for the authors and I do not require further review of this paper. The paper can be published after addressing those comments.

(1) I read the explanation given by the authors regarding how to decide the size of the interface model. It seems that the length was decided arbitrarily, and the number of ions was decided by ensuring the model shows EDL features and bulk phase structures. However, whether the simulated system is in equilibrium under a chosen temperature and pressure is not known? The density used to decide the number of each electrolyte species for MD simulation is an experimental result, not a simulation result, checking the equilibrium state is necessary when using an experimental density.

(2) The method to calculate voltage (potential vs PZC) based on the equation given in the rebuttal letter is questionable. The way to calculate potential difference with the constant charge model is normally by solving Poisson's equation ($d^2\psi/dz^2 = -\rho/\epsilon_0$) in the Z-direction, and this should be done for both charged and uncharged electrode models to decide the surface charge density for -1 V vs. PZC. One example can be found in J. Phys. Chem. C 2019, 123, 3, 1610–1618. However, if the authors do not emphasize the simulated voltage of -1 V vs. PZC, but only consider EDL near a negatively charged electrode surface with the applied surface charge density given clearly, I think the simulation method is acceptable.

REVIEWERS' COMMENTS

Reviewer #2 (Remarks to the Author):

I only have two comments for the authors and I do not require further review of this paper. the paper can be published after addressing those comments.

(1) I read the explanation given by the authors regarding how to decide the size of the interface model. It seems that the length was decided arbitrarily, and the number of ions was decided by ensuring the model shows EDL features and bulk phase structures. However, whether the simulated system is in equilibrium under a chosen temperature and pressure is not known? The density used to decide the number of each electrolyte species for MD simulation is an experimental result, not a simulation result, checking the equilibrium state is necessary when using an experimental density.

Response:

We thank Reviewer #2 for the recommendation to publish our paper, as well as the very useful and in-depth comments on the system equilibrium state. The remaining comments are addressed below, with indication of the changes in the manuscript and the SI.

Since systematically optimizing the length of two electrodes is out of the scope of this study, we intentionally chose a length (10 nm) that is large enough for eliminating the electrostatic interaction force of charged electrodes on the bulk electrolyte molecules/ions. As we expected, in our simulation, the remaining force was indeed eliminated in the bulk electrolyte and simulation results provided reasonable electrolyte bulk solvation structures which were cross-verified with Raman spectroscopy vs RDF. We, therefore, believe the 10 nm electrode distance provided a reliable simulation condition.

Using this 10 nm electrode distance condition, we added the number of ions/molecules corresponding to the real experimental electrolyte density under a chosen temperature and pressure. As required by Reviewer#2, the density of electrolytes under equilibrium conditions has been checked to compare with the experimental value as shown in Figure 1. After NVT equilibrium, we then performed the NPT equilibrium for the simulation system using the same number of ions/molecules at room temperature (300 K), and the pressure was controlled at around 1 bar (Figs. 1a, b). The equilibrium state of the system was checked by monitoring the density fluctuations of the electrolyte during the 100 ps equilibrium simulation period (Figs. 1d, e) [1]. According to the density fluctuation curves, the averaged density of EC-based electrolyte (1305 kg/m³) and DX-based electrolyte (1079 kg/m³) after equilibrium are close to the experimental values with acceptable error (~2 %) and (~8 %), respectively, which indicates that the density values we chose for the simulation can guarantee the simulation systems are an equilibrium state.

This equilibrium state checking was added in **Supplementary Figs. 18c-f**, with the description in **Lines 18-21, Page 21** of the revised **supplementary information**.

Figure 1 (a, b) Pressure fluctuation curves and (c, d) density fluctuation curves of EC- and DX-based electrolyte during the equilibrium simulation.

(2) The method to calculate voltage (potential vs PZC) based on the equation given in the rebuttal letter is questionable. The way to calculate potential difference with the constant charge model is normally by solving Poisson's equation ($d^2\Psi/dz^2 = -\rho/\epsilon_0$) in the Z-direction, and this should be done for both charged and uncharged electrode models to decide the surface charge density for -1 V vs. PZC. One example can be found in J. Phys. Chem. C 2019, 123, 3, 1610–1618. However, if the authors do not emphasize the simulated voltage of -1 V vs. PZC, but only consider EDL near a negatively charged electrode surface with the applied surface charge density given clearly, I think the simulation method is acceptable.

Response:

We thank the Reviewer for the to very relevant suggestion. The estimation of the number of charges added to the carbon atoms in the electrode was obtained from Ref. [2] below that was **added as a Ref 98 in the revised manuscript** which used the similar distance of two electrodes and the same electrolyte (1 M LiPF₆ in EC and DMC, with minor changes in the solvent ratio). According to Ref. 2, our charge density (18.42 $\mu\text{C}/\text{cm}^2$) applied to the electrode in the EC-based electrolyte corresponds to a voltage value of around a few volts, which is close to the real experiment value. Moreover, since our electrolyte system has standard molarity, the voltage drops almost linearly inside the simulation box, similar to a parallel capacitance in which the voltage dropping most occurs in the bulk electrolyte, not the electrode surface. Therefore, our previous estimation method could give a reasonable proximation of electrode voltage. We also agree with Reviewer #2 that a more precise determination of electrode voltage should be done by solving Poisson's equation as specified in Ref 1 and Ref 3 (suggested

by the Reviwer and added as a **Ref 99 in therevised manuscript**). As suggested by Reviewer #2, in the revised manuscript we do not emphasize the exact value of the simulation voltage of -1 V vs PZC, that is not relevant to the electrical double layer structure but instead point out the applied negative surface charge on the carbon electrode. These changes are applied in **Lines 4-6, Page 14 in the revised main manuscript**.

We also completed all requested formatting changes in the manuscript and SI that are indicated in the author checklist.

We would like to thank again the Reviwers and the Editorial board for the excellent comments and guidance that allowed to significantly improve the original manuscript. The changes stated in the the manuscript and the SI are highlighted, with non-highlighted version are also provided.

References:

- [1] Xing, L., Vatamanu, J., Borodin, O., Smith, G. D. & Bedrov, D. Electrode/Electrolyte Interface in Sulfolane-Based Electrolytes for Li Ion Batteries: A Molecular Dynamics Simulation Study. *The Journal of Physical Chemistry C* **116**, 23871-23881, doi: 10.1021/jp3054179 (2012).
- [2] Boyer, M. J., Vilciauskas, L., & Hwang, G. S. Structure and Li⁺ Transport in a Mixed Carbonate/LiPF₆ Electrolyte Near Graphite Electrode Surface: A Molecular Dynamics Study. *Phys. Chem. Chem. Phys.* **18**, 27868, doi: 10.1039/c6cp05140e (2016)
- [3] Fang, A., & Smolyanitsky, A. Simulation Study of the Capacitance and Charging Mechanisms of Ionic Liquid Mixtures near Carbon Electrodes. *J. Phys. Chem. C* **123**, 1610–1618, doi: 10.1021/acs.jpcc.8b10334 (2019)